# Regionalization with Hierarchical Hydrologic Similarity and Ex-situ Data for the Estimation of Mean Annual Groundwater Recharge at Ungauged Watersheds

Ching-Fu Chang[1] and Yoram Rubin[1]

[1]Department of Civil and Environmental Engineering, University of California, Berkeley, U.S.A.

**Correspondence:** Yoram Rubin (rubin@ce.berkeley.edu)

**Abstract.** There are various methods available for annual groundwater recharge estimation with in-situ observations (i.e., observations obtained at the site/location of interest), but a great number of watersheds around the world still remain ungauged, i.e., without in-situ observations of hydrologic responses. One approach for making estimates at ungauged watersheds is regionalization, namely, transferring information obtained at gauged watersheds to ungauged ones. The reliability of regionalization depends on (1) the underlying system of hydrologic similarity, i.e., the similarity in how watersheds respond to precipitation input, as well as (2) the approach by which information is transferred.

In this paper, we present a nested tree-based modeling approach for conditioning estimates of hydrologic responses at ungauged watersheds on ex-situ data (i.e., data obtained at sites/locations other than the site/location of interest) while accounting for the uncertainties of the model parameters as well as the model structure. The approach is then integrated with a hypothesis of two-leveled hierarchical hydrologic similarity, where the higher level determines the relative importance of various watershed characteristics under different conditions, and the lower level performs the regionalization and estimation of the hydrologic response of interest.

We apply the nested tree-based modeling approach to investigate the complicated relationship between mean annual groundwater recharge and watershed characteristics in a case study, and apply the hypothesis of hierarchical hydrologic similarity to explain the behavior of a dynamic hydrologic similarity system. Our findings reveal the decisive roles of soil available water content and aridity in hydrologic similarity at the regional and annual scales, as well as certain conditions under which it is risky to resort to climate variables for determining hydrologic similarity. These findings contribute to the understanding of the physical principles governing robust information transfer.

## 1 Introduction

Groundwater resources supply approximately 50% of the drinking water and roughly 40% of the irrigation water worldwide (National Ground Water Association, 2016). Yet, the groundwater has increasingly been depleted since the late 20th century (Wada et al., 2010). Therefore, groundwater recharge, here broadly defined as the replenishing of water to a groundwater reservoir, plays a critical role in sustainable water resources management (de Vries and Simmers, 2002). Several studies have reviewed and compared multiple methods for recharge estimation at a wide spectrum of temporal and spatial scales, including

lysimeter tests, seepage tests, water table fluctuation, chemical and heat tracers, baseflow analysis, water budget, and numerical modeling (Scanlon et al., 2002; Healy, 2010; Heppner et al., 2007). However, the aforementioned methods rely on in-situ data, while many watersheds worldwide still remain effectively ungauged (i.e., ungauged, poorly gauged, or previously gauged) (Loukas and Vasiliades, 2014).

This fact leads us to a critical question: How can one estimate hydrologic responses without in-situ data? Studying ungauged watersheds has been a popular research topic for more than a decade, especially since The Prediction in Ungauged Basins (PUB) initiative by the International Association of Hydrological Sciences (IAHS) (Sivapalan et al., 2003). Facing the lack of in-situ data, studies have attempted transferring ex-situ information from gauged watersheds to ungauged ones; this data transfer is also termed "regionalization". Regionalization has been applied to constrain the estimates of the parameters of

hydrologic models (especially rainfall-runoff models), which could then be used to make predictions at ungauged watersheds (Kuczera, 1982; Singh et al., 2014; Razavi and Coulibaly, 2017; Wagener and Montanari, 2011; Blöschl et al., 2013). Such constraining is expected to lead to more accurate and precise estimates, and could be in the form of (1) relationships between model parameters and watershed characteristics, (2) subsets of the parameter space, or (3) plausible parameter values from models built for other hydrologically similar watersheds (Singh et al., 2014).

However, the application of regionalization is not without challenges. One of the key factors of predictive uncertainty identified by the PUB initiative is the unsuitability of information transfer techniques, due to a lack of comparative studies across watersheds and a lack of understanding of the physical principles governing robust regionalization (Hrachowitz et al., 2013). Different regionalization techniques have been applied in different cases with different assumptions. For example, Li et al. (2018) attempted a simple form of regionalization, where kernel density estimation was applied on recharge values obtained

from various hydrologically similar sites, in order to build an ex-situ prior distribution (i.e., a prior distribution conditioned on ex-situ data). However, one limitation in Li et al. (2018) was that hydrologic similarity was treated as a Boolean variable, and therefore, there was no way to systematically distinguish a highly similar site from a slightly similar site. To pursue this further in this study, we must ask the following question: How can we tell that two watersheds are hydrologically similar? Sawicz et al. (2011) applied Bayesian mixture clustering to watersheds across the eastern U.S. They found that spatial proximity

was a valuable first indicator of hydrological similarity because it reflected strong climatic control in their study area. Oudin et al. (2008) reported similar findings based on 913 French watersheds, despite acknowledging the lack of some key physical descriptors in their data set. However, Smith et al. (2014) attempted regionalization of hydrologic model parameters in eastern Australia, and suggested that spatial proximity was an unreliable metric of hydrological similarity. For their part, Tague et al. (2013) presented successful regionalization of hydrologic parameters based on geologic similarity at watersheds in the U.S.

Oregon Cascades, a mountain range that features geological heterogeneity. Although not directly shown, their findings also went against the use of applying spatial proximity, for they discussed the sharp contrasts in hydrology at proximal watersheds based primarily on geological differences. The indication from these findings is that, although spatial proximity is of practical importance due to its common use, its simplicity, and its demonstrated effectiveness in specific areas (Smith et al., 2014), it is not the true controlling factor, but rather a confounding factor.

One can resort to other physical characteristics of watersheds for the determination of hydrologic similarity. However, what those characteristics are may be a complicated question. Razavi and Coulibaly (2017) tested the effect of combinations of neural-network-based classification techniques and regionalization techniques in Canada, and found that classifying watersheds before regionalization improves regionalization for streamflow, baseflow, and peak flow predictions, but also discovered that the best combination of techniques varied from one watershed to another. Singh et al. (2014) applied classification and regression tree to determine the relationship between catchment similarity and regionalization in the U.S., finding that the dominant controls of successful regionalization vary significantly with the spatial scale, with the region of interest, and with the objective function used. Similarly, Kuentz et al. (2017) found that different physiographic variables controlled various flow characteristics across Europe, showing how different descriptors could account for different dominant hydrologic processes and flow characteristics. These studies indicate an important challenge, that the factors determining hydrologic similarity may vary under different conditions, and a universal system of hydrologic similarity still remains unavailable. Loritz et al. (2018) suggested an interesting perspective describing a dynamic hydrologic similarity system, where similarity and uniqueness are not mutually exclusive; rather, they suggested that hydrologic systems operate by gradually changing to different levels of organization in which their behaviors are partly unique and partly similar.

In this study, we would like to integrate the perspective in Loritz et al. (2018), that similarity and uniqueness are not mutually exclusive, into our regionalization framework for groundwater recharge estimation at ungauged watersheds. It is thus critical to identify a number of plausible controlling factors. Although few studies have directly identified the controlling factors, some insights can be learned from previous studies. For example, the effective recharge (i.e., the net source term in the groundwater flow equation) in a steady, depth-integrated, and unbounded groundwater flow was found to be correlated with the spatial distributions of transmissivity and hydraulic head (Rubin and Dagan, 1987a, b). From a recharge-mechanism-based perspective, previous studies have also found a list of plausible controlling factors of recharge via recharge potential mapping (Yeh et al., 2016, 2009; Naghibi et al., 2015; Rahmati et al., 2016). These variables include watershed topography, land cover, soil properties, and geology. At the regional scale, climate variables have been found to be among the primary controlling factors of groundwater table depth (Fan et al., 2013), mean annual groundwater recharge (Nolan et al., 2007), and mean annual baseflow (Rumsey et al., 2015), the latter of which is often used as a surrogate of recharge under the steady state assumption. Other examples include Xie et al. (2017), who showed that evapotranspiration data provided more conditioning power and more uncertainty reduction than soil moisture data in long-term mean recharge estimation, and Hartmann et al. (2017), who reported variations of the sensitivity of annual groundwater recharge to annual precipitation with aridity. Although these studies did not apply regionalization explicitly and did not target ungauged watersheds directly, their findings provide guidance for us to identify some watershed characteristics—especially climate variables—that might play an important role in the regionalization process for recharge estimation.

Given a set of watershed characteristics, the next important question is how the regionalization is carried out. Gibbs et al. (2012) provided a generic framework of regression regionalization, which involves a multi-objective optimization for calibration, a sensitivity analysis to determine the most important model parameters, and a final step relating watershed characteristics with model parameters. The framework is capable of assimilating information from exogenous variables and accounting for

the interaction between parameters. However, the framework does not include a straightforward quantification of uncertainties in calibration and in regionalization. In comparison, Bayesian approaches offer a solution to the quantification of uncertainty by outputting conditional distributions. Despite the lack of in-situ data, one can still apply Bayesian approaches to establish prior distributions that are informed by data from previous studies or well-established databases (Woodbury and Rubin, 2000; Hou and Rubin, 2005; Woodbury, 2011). More advanced pooling of information from multiple sampled sites has also been demonstrated with the application of Bayesian hierarchical models (Smith et al., 2014; Cucchi et al., 2019), which can account for both intra- and inter-site uncertainty of the parameters. However, the aforementioned Bayesian approaches have several disadvantages, including: (1) requiring a system of hydrologic similarity that helps us decide which sampled sites or databases are suitable as "information donor", (2) requiring known or assumed distributional forms of the parameters, and (3) difficulties in accounting for complicated and highly non-linear dependence on exogenous variables. Adding onto the challenge is that uncertainty arises from a lack of knowledge about how to represent the watershed system in terms of both model structure and parameters (Beven, 2016). Uncertainty about the model structure has been identified and studied, (e.g., Beven, 2006; Beven and Freer, 2001; Nowak et al., 2010), but not under the context of ungauged watershed, regionalization, and hydrologic similarity. The lack of in-situ data does not justify a presumed model structure; even without in-situ data, the modeler can still consider simultaneously multiple potential model structures, instead of wrongly assuming a fixed structure (Rubin et al., 2018).

To that end, the objectives of this study are twofold. First, to address the aforementioned challenges in regionalization technique, we propose a nested tree-based modeling approach, which features (1) non-linear regression in order to model the predictor-response relationship, (2) full Bayesian quantification of parameter uncertainty, and (3) proposal-comparison-based consideration of model structure uncertainty. Second, we integrate the nested tree-based modeling approach with a hypothesis of hierarchical hydrologic similarity. We apply the approach to estimate mean annual groundwater recharge at ungauged watersheds in a case study, and we invoke the hypothesis of hierarchical similarity to reveal the key controlling factors of a dynamic hydrologic similarity system, which could ultimately contribute to robust information transfer in future applications.

## 2  Methodology

The data-driven, Bayesian, and non-linear regression approach proposed in this study is powered by Bayesian Additive Regression Tree (BART) at its core. The details of BART, including the establishment of prior distribution (which we term prior), the calculation of likelihoods, and the posterior inference statistics are well documented in Chipman et al. (2010) and in Kapelner and Bleich (2016). Here, we provide a brief conceptual introduction to the implementation and advantages of BART, as well as how BART it augmented in this study.

## 2.1 BART

Consider a fundamental problem of making inference about an unknown function that estimates a response variable of interest using a set of predictor variables. The general form of this problem can be expressed as follows:

$$R = \hat{R} + \epsilon = f(\boldsymbol{\theta}, \mathbf{x}) + \epsilon, \tag{1}$$

where $R$ is the response variable, $f(\cdot)$ is a model that outputs the estimate of the response variable, $\hat{R}$ is the estimate, $\boldsymbol{\theta}$ is the vector of model parameters, $\mathbf{x}$ is the vector of predictors, and $\epsilon$ is a Gaussian white noise with finite variance, i.e., $\epsilon \sim N(0, \sigma^2)$. The observation of $R$ is denoted by $r$. BART solves this problem by applying a Bayesian version of the additive ensemble tree model. To put it simply, BART can be understood as Bayesian inference done for many individual regression tree models. The main difference between typical regression tree models and BART is that the former is calibrated with data by searching for the best model parameters that lead to the least error, while the latter is conditioned on data by obtaining conditional distributions of model parameters via Bayesian inference.

To understand BART, first one needs to understand the build-up of the additive ensemble tree model from individual Classification and Regression Tree (CART) models (Breiman, 1984). A schematic diagram of a CART model is shown in Fig. 1(**a**), which resembles an upside-down tree (root on top and leaves at the bottom). The root node of the tree represents the space spanned by the predictor(s). As one moves downward from root to leaves, the said space is recursively partitioned by a sequence of binary partitioning rules. This partitioning and the corresponding partitioning rules define the tree structure, and can be represented by the tree structure variable, denoted by $T$. After partitioning, output response values are assigned to each and every leaf, where each leaf represents a partitioned subspace. These output values can be collectively denoted by $\mathbf{M}$. A tree model can be fully defined by knowing its $T$ and $\mathbf{M}$.

To further improve the predictive performance on an individual CART, an additive ensemble tree model can be built as the sum of $J$ individual trees (Fig. 1(**b**)), each of which has its tree structure ($T_j, j = 1, ..., J$) and its set of leaf values ($\mathbf{M}_j, j = 1, ..., J$), shown as follows:

$$\hat{R} = f(\boldsymbol{\theta}, \mathbf{x}) = \sum_{j=1}^{J} g(T_j, \mathbf{M}_j, \mathbf{x}). \tag{2}$$

where $\boldsymbol{\theta} = \{T_1, \mathbf{M}_1, ..., T_J, \mathbf{M}_J\}$ and $g(\cdot)$ denotes an individual tree. The output of an additive ensemble tree model is the sum of the outputs from the $J$ trees.

Like mentioned above, instead of searching for the best $T_j$ and $\mathbf{M}_j$ for every $j$ that lead to the least error, BART takes on a different way of model fitting, the Bayesian way. It starts by defining the following joint prior of all the tree structures, all the sets of leaf values, and the variance of the white noise defined in Eq. 1:

$$p\left(T_1, \mathbf{M}_1, ..., T_J, \mathbf{M}_J, \sigma^2\right) = p(\sigma^2) \prod_{j=1}^{J} p(T_j) P(\mathbf{M}_j | T_j). \tag{3}$$

BART then applies a tailored version of backfitting Markov Chain Monte Carlo (MCMC) simulation algorithm to condition the prior on the response data ($r$), where backfitting means the $j$th tree model is iteratively updated with its partial residual. The

stationary distribution toward which the MCMC simulations converge is then used to approximate the true posterior distribution (which we term posterior):

$$p\left(T_1, \mathbf{M}_1, ..., T_j, \mathbf{M}_j, \sigma^2 | r\right). \tag{4}$$

A schematic diagram of the MCMC simulation iteration procedure is shown in Fig. 1(c). Within each MCMC simulation, both $T_j$ and $\mathbf{M}_j$ for the $j$th tree are iteratively simulated using a Metropolis-within-Gibbs sampler, illustrated by the loop in the blue circle in Fig. 1(c). After simulating all the trees, the error variance ($\sigma^2$) is simulated with a Gaussian-Gamma-conjugate Gibbs sampler. The sampling of $\sigma^2$ marks the end of one MCMC simulation. We can see by the loop in the red square in Fig. 1(c), the MCMC simulation is continuous, until the simulated values converge to a stationary distribution. These post-convergence simulated values approximate realizations from Eq. 4, and thus we approximate the true posterior in Eq. 4 by the stationary distribution obtained by MCMC simulation. At this point, we have reached a BART model that is conditioned on the response data, because all the BART parameters (tree structures, leaf node values, and the white noise variance) have been conditioned on the response data.

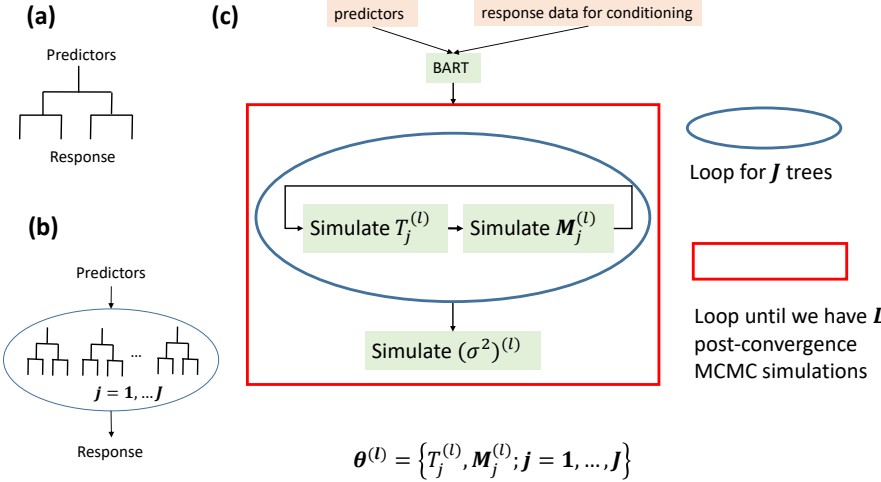

**Figure 1.** Schematic diagrams of (a) a regression tree model, (b) an ensemble tree model which consists of $J$ additive regression tree models, and (c) the loops structure that BART uses to draw MCMC simulations (indexed by $l$), consisting of an inner loop for $J$ additive regression tree models and an outer loop that continues until we have a total of $L$ MCMC simulations after convergence toward a stationary distribution.

Given the aforementioned conditioned BART model, we now turn our attention to estimating a new response that was not included in the data on which the BART model was conditioned. This is done by inputting the vector of the new predictors, denoted by $\tilde{x}$, into the predictor-response relationship we learned with the BART model. Firstly, Eq. 1 can be rewritten as:

$$R \sim N\left(\hat{R}, \sigma^2\right). \tag{5}$$

Both the mean and the variance in Eq. 5 are uncertain, and have their respective posteriors. By combining Eqs. 2 and 5, and after plugging in the post-convergence MCMC simulated values and $\tilde{\mathbf{x}}$, we obtain a plausible realization (indexed by the superscript $l$, $l = 1, ..., L$) of predictive distribution as follows:

$$
N\left(\hat{R}^{(l)}, (\sigma^2)^{(l)}\right) =
$$
$$
N\left(f\left(\boldsymbol{\theta}^{(l)}, \tilde{\mathbf{x}}\right), (\sigma^2)^{(l)}\right) =
$$
$$
N\left(\sum_{j=1}^{J} g\left(T_j^{(l)}, \mathbf{M}_j^{(l)}, \tilde{\mathbf{x}}\right), (\sigma^2)^{(l)}\right). \tag{6}
$$

The collection of many plausible realizations yields an approximated posterior of predictive distributions. Thus, for response of interest, we have now obtained a fully Bayesian Gaussian predictive model, where the mean and the variance have their respective posteriors.

## 2.2 Advantages of BART

The key advantage of BART is that it combines the non-linear regression for the predictor-response relationship with Bayesian inference, allowing for the determination of a full Bayesian posterior of predictive distribution, rather than one or a few estimates/predictions.

The estimation and regionalization processes are data-driven. Prior knowledge of the underlying physics is only minimally accounted for in terms of the composition of the predictor sets and the user-defined prior of the splitting rules (which are embedded in the tree structure variable, $T_j$). The underlying physics is inferred from the ex-situ data via obtaining conditional simulations of the tree structures and the leaf nodes (similar to the calibration stage), and thus, is implicitly embedded rather than explicitly defined. Therefore, the extent to which physics could be inferred is restricted by the training data —here, the ex-situ data, which is a common limitation of data-driven approaches.

However, in compensation, we avoid one disadvantage of the application of physically based models in the case of ungauged watersheds. The available data at the ungauged watershed are limited, and it is unrealistic to expect that certain watershed characteristics should be known. Data availability could hinder the implementation of powerful hydrologic models (Razavi and Coulibaly, 2017) because some of the required model inputs may be unavailable at the ungauged watersheds (Xie et al., 2017; Gemitzi et al., 2017). It is possible to treat missing inputs as parameters, and run simulations to impute them or apply stochastic methods to estimate them. Nonetheless, the corresponding computational demand grows in power law with the number and the plausible range of the missing inputs, which is of great practical importance when evaluating the pros and cons of an approach.

Note that in this study there is no intention to show the superiority of either the data-driven or the physically based approaches. As Wagener and Montanari (2011) pointed out, the ultimate goal of predictions at ungauged watersheds is not to define parameters of a model, but rather, to understand what behavior we should expect at the ungauged watersheds of interest. We have simply shown why our approach is suitable for ungauged watersheds.

## 2.3 Nested tree-based modeling approach

Like shown above, BART offers an elegant way to account for model parameter uncertainty of an additive ensemble tree model. However, uncertainty exists not only for the model parameters but also for the models themselves, i.e., the model structure uncertainty. A significant factor of model structure uncertainty for BART could be the composition of the vector of predictors. Accounting for model structure uncertainty can be done by proposing a prior probability mass function of plausible BART models, which can then be evaluated and compared with each other. In the present study, we accomplish this by using a proposal-comparison procedure, which we termed the nested tree-based modeling approach. The details are as follows.

We start by proposing $K$ plausible BART models, denoted as $B_k, k = 1, ..., K$, each of which is built using a unique set of predictors and is conditioned on available data. The model structure uncertainty is accounted for by obtaining a probability mass function of the $K$ plausible BART models, denoted by $p(B_k)$. The determination of $p(B_k)$ can be informed by the data (namely, in an empirical Bayes way, where the prior is informed by the data). At each available data point, we evaluate the performance of the plausible BART models by a performance metric (a typical example is the mean squared error). Then, a label is given to each data point, indicating which BART model has the highest performance measured by the metric. Finally, we use a CART model to classify the data points based on their labels. The CART model outputs an empirical multinomial distribution of the $K$ plausible BART models at each leaf. Thus, one can study the variation of $p(B_K)$ with various predictors. A very simple example is illustrated in Fig. 2, where we compare the performances of two BART models ($K = 2$) using one predictor and a simple two-leveled classification tree. The predictor space is partitioned into the positive subspace and the negative subspace by the partitioning rule indicated in the diamond box. Thus, for any new data point with positive predictor value, we would use $p(B_1) = 0.76$ and $p(B_2) = 0.24$ as the probability mass function of plausible models. In real applications, of course, one can use an arbitrary number of predictors to compare an arbitrary number of plausible BART models.

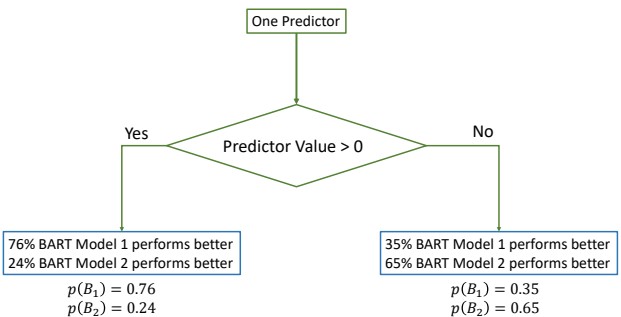

**Figure 2.** Schematic diagrams of an example of nesting two BART models under a simple two-leveled CART model, using only one predictor. The partitioning rule is expressed in the diamond box, and the leaves are represented in blue boxes.

Up to this point, we have introduced the nested tree-based modeling approach, which is general and data-driven. For estimation purpose, one would be interested in accounting for model structure uncertainty by averaging the estimates over $p(B_k)$,

which can be done by invoking Bayesian model averaging. However, the capability of the nested tree-based modeling approach does not stop here, as the approach also outputs the variation of $p(B_k)$ under various conditions. This could be an indication of the behavior of a dynamic hydrologic similarity system, and will be explained in details in Sect. 2.4.

## 2.4 Hypothesis of hierarchical similarity

To facilitate the interpretation of the variation of $p(B_k)$, we propose a hypothesis of hierarchical similarity that has two levels:

1. The lower level is termed the **predictor similarity**, meaning that if two vectors of predictors are similar in some parts, their corresponding response will be similar. In hydrology context, if two watersheds have some similar characteristics, then their hydrologic responses will be similar. This lower level corresponds to the BART models in the nested tree-based modeling approach.

2. The higher level is the **regionalization similarity**, meaning that if two vectors of predictors are similar in some parts, their corresponding predictor-response relationships will be similarly controlled. In hydrology context, if two watersheds have some similar characteristics, then their hydrologic responses will be governed by similar functions/mechanisms. This higher level corresponds to the classification tree in the nested tree-based modeling approach.

Put simply, regionalization similarity determines the predictor-predictor relationship and tells us which predictors to extract information from, while predictor similarity determines the predictor-response relationship that actually estimates the response using the said extracted information. Note that the two sets of predictors respectively determining the two levels of similarity are not mutually exclusive: they may or may not overlap. To elaborate on the difference between the two levels of similarity, we present the following two example statements within the context of recharge estimation.

1. **Systematic trends in recharge rates are often associated with climatic trends (Healy, 2010).** This is a statement of predictor similarity, indicating a predictor-response relationship. One would be informed to association recharge rates with climatic variables.

2. **In arid regions, focused recharge from ephemeral streams is often the dominant form of recharge (Healy, 2010).** This is a statement of regionalization similarity, indicating a predictor-predictor relationship. One would be informed to pay more attention to the dominant factors of ephemeral streams, if the study area of interest is in arid regions.

Having explained the hypothesis of hierarchical similarity, now suppose that we have gone through the process described in Sect. 2.3, and have obtained $K$ plausible BART models and one CART model. Each plausible BART model was built with a unique set of predictors, and we use the BART models to explore predictor similarity with different predictor sets. Moving up a level, we use the classification tree to explore regionalization similarity by investigating the variation of $p(B_k)$ under various conditions. Note that as the condition changes, the best performing BART model may change and so does the set of dominant predictors in the predictor-response relationship. This may explain why under different conditions, the hydrologic similarity may be controlled by different watershed characteristics. We test our hypothesis of hierarchical similarity in a case study, which will be explained in Sect. 3.

# 3   Case study

In this case study, we are going to apply the methodology described in Sects. 2.1 through 2.4 to investigate the predictor similarity and the regionalization similarity in the study area, and to test the hypothesis of hierarchical similarity. It is important to note that this case study is not aimed at a thorough investigation of the recharge mechanism, nor is the goal obtaining the most accurate recharge estimates. Rather, the primary goals are the demonstration of the power of our approach, and showing how the approach helps us understand the dynamic behavior of hydrologic similarity in the study area. This Sect. provides the details about the case study setup, including the watersheds, the recharge data, the watershed characteristics data, the partitioning of data, and the evaluation metrics.

## 3.1   Watersheds and recharge estimates

The conterminous United States can be divided into eight major river basins (MRBs), each of which consists of thousands of watersheds (The United States Geological Survey, 2005; Brakebill and Terziotti, 2011). At each and every watershed, watershed-average annual recharge estimate and watershed characteristics data are retrieved from publicly available databases, and will be described in the following subsections. In our work, the recharge estimates are used as the target response while the characteristics are used as predictors in the regionalization process.

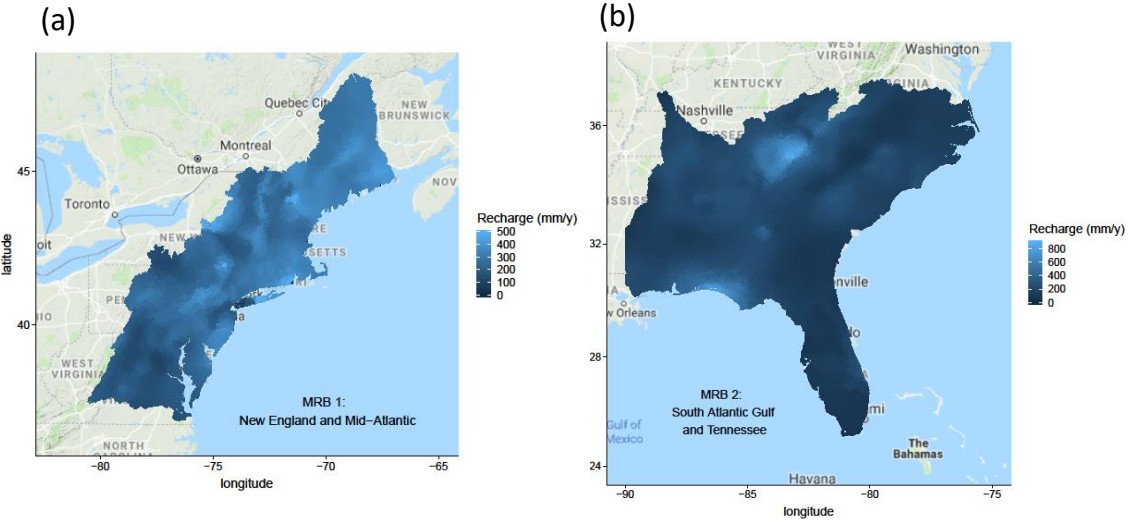

**Figure 3.** The study area includes **(a)** MRB 1 and **(b)** MRB 2 in the eastern U.S., colored by the estimated annual groundwater recharge in the year of 2002 (Wolock, 2003). For the details of the delineation of MRBs please refer to The United States Geological Survey (2005).

In 2002, annual groundwater recharge at each watershed was estimated via baseflow analyses by the U.S. Geological Survey (USGS) (Wieczorek and LaMotte, 2010h; Wolock, 2003, also shown in Fig. 3). Streamflow-based estimation of recharge, such as baseflow analysis, is commonly used in humid regions. As put forward by Healy (2010), there are three key questions that should be carefully checked before applying baseflow analysis: (1) Is all recharging water eventually discharged into the stream where the baseflow is measured? (2) Do low flows consist entirely of groundwater discharge? (3) Does the contributing area of the aquifer differ significantly from that of the watershed? Without a rigorous proof, we make a working assumption about the reliability of baseflow analysis. Fortunately, from a post hoc check, the recharge estimates fall within the typical scales at which baseflow analysis is more suitable: a recharge scale from hundreds to thousands $mm$ per year, a spatial scale of hundreds of $m^2$ to hundreds of $km^2$, and temporal scales from months to decades (Scanlon et al., 2002).

The more arid U.S. Midwest may have more pronounced localized recharge (de Vries and Simmers, 2002), which cannot be effectively captured by baseflow analysis (Scanlon et al., 2002). This, then, does not fit well with our working assumption. Therefore, following the suggestion of Nolan et al. (2007), our study area includes only the relatively humid eastern parts of the U.S., namely MRB 1 and 2 (Fig. 3). After excluding watersheds with less desirable data coverage, we consider a total of 3609 watersheds in MRB 1 and 7413 watersheds in MRB 2. The distributions of the recharge data from all the watersheds in the study area are shown in Fig. 4**(a)**.

## 3.2  Climate

At each watershed included in the study, the following data are retrieved from publicly available databases: the long-term average annual precipitation ($\bar{P}$) averaged from 1970 to 2000 (Wieczorek and LaMotte, 2010a), the annual precipitation in the year 2002 ($P$) (Wieczorek and LaMotte, 2010b), and the long-term average annual potential evapotranspiration ($E_p$) averaged from 1960 to 1990 (Title and Bemmels, 2017). Note that limited by data availability, the average periods of $\bar{P}$ and $E_p$ are different. Thus, we also make a working assumption that at the decadal scale the averaged climate variables remain steady, with which we ignore the potential effect of climate change on the difference between the average from 1960 to 1990 and that from 1970 to 2000. Given the precipitation and evapotranspiration, we obtained two additional climate variables: the long-term aridity index, estimated as $\bar{\phi} = E_p/\bar{P}$, and the 2002 aridity index, estimated as $\phi = E_p/P$. Given that the recharge data are based on baseflow analysis for the year 2002, $P$ and $\phi$ represent the climate controls of that same year, while $\bar{P}$, $E_p$, and $\bar{\phi}$ represent climate controls over the long-term. The distributions of $P$, $\bar{P}$, and $E_p$ are shown in Figs. 4**(b)**, **(c)**, and **(d)**, respectively.

### 3.2.1  Normalization and transformation of recharge using precipitation

The annual recharge data (in volume of water per unit watershed area) can be normalized by $P$ (also in volume of water per unit watershed area), as in Fig. 4**(e)**. This stems from the concept of water budgets and has been commonly used in hydrological studies worldwide (e.g., Magruder et al., 2009; Rangarajan and Athavale, 2000; Obuobie et al., 2012; Heppner et al., 2007; Takagi, 2013; Yang et al., 2009). Here, we apply logit transformation, which is common for proportions or probabilities (Gelman et al., 2014), to that normalized recharge, relaxing the physical bounds (0 and 1) of the values of the target variable

(Fig. 4(**f**)). This step is advantageous as it opens the opportunity to estimate recharge with parametric statistical models without special accommodations for the bounds. Therefore, in this case study the logit normalized recharge (LNR) is used as the target response variable.

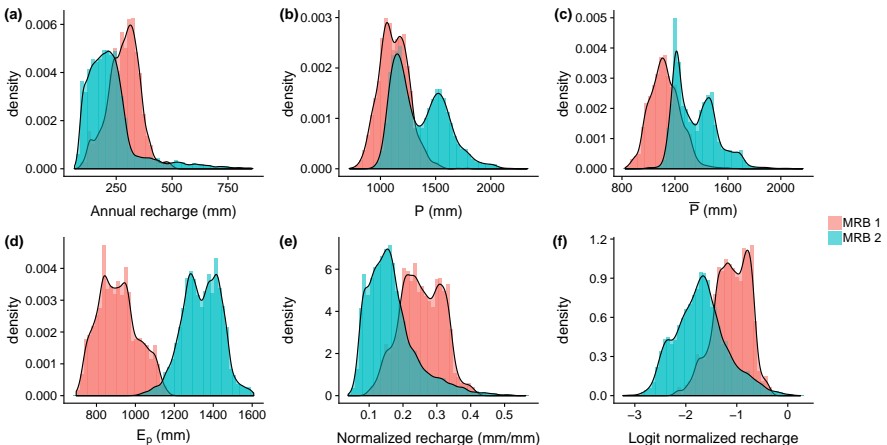

**Figure 4.** Histograms of (**a**) annual recharge in 2002, (**b**) annual precipitation in 2002, (**c**) long term average annual precipitation, (**d**) long term average annual potential evapotranspiration, (**e**) normalized recharge, and (**f**) logit normalized recharge (LNR) at all the watersheds in MRB 1 and 2. The black curves are estimates of the distributions based on kernel density estimation.

## 3.3 Non-climate watershed characteristics

We also consider various non-climate watershed characteristics in this study, including topography, land cover, soil properties, and geology. The land cover is based on data published in 2001, which we feel is close enough to 2002 to provide the appropriate information. The other characteristics are based on raw data obtained in different years before 2002; it is assumed that they remain steady at sub-century time scales. We provide the details of these watershed characteristics in the following subsections.

### 3.3.1 Topography and land cover

The topographic predictors are taken from publicly available databases (Wieczorek and LaMotte, 2010g); they are summarized in Table 1. The land cover variables are the percentages of watershed area corresponding to each land cover class (Wieczorek and LaMotte, 2010f); these are summarized in Table 2. The land cover classes are based on the 2001 National Land Cover Database (NLCD2001), the categories of which include water, developed land, barren land, forest, shrubland, herbaceous land, cultivated land, and wetland, with each having its own sub-classes. The details of NLCD2001 can be found in Homer et al. (2007).

**Table 1.** Watershed topography predictors.

| Variable | Explanation |
|---|---|
| Basin index | Watershed area divided by watershed perimeter squared (dimensionless). |
| Stream density | Reach length divided by watershed area ($m^{-1}$). |
| Sinuosity | Reach length divided by the length of the straight line connecting the beginning and the ending of the reach (dimensionless). |
| Slope | Mean watershed slope calculated from digital elevation data (degree). |

**Table 2.** Land cover classification by NLCD2001.

| Class | Subclass |
|---|---|
| Water | Open water |
| | Perennial ice |
| Developed | Open space |
| | Low intensity |
| | Medium intensity |
| | High intensity |
| Barren | Barren land |
| Forest | Deciduous |
| | Evergreen |
| | Mixed |
| Shrubland | Dwarf shrub |
| | Shrub/scrub |
| Herbaceous | Grassland |
| | Sedge |
| | Lichens |
| | Moss |
| Cultivated | Pasture/hay |
| | Crops |
| Wetlands | Woody wetland |
| | Emergent herbaceous wetland |

## 3.3.2 Soil property

The soil property predictors include watershed scale statistics (e.g., average, upper bound, and lower bound) of soil properties (Wieczorek and LaMotte, 2010e); these are summarized in Table 3. The spatial statistics of the soil properties within each watershed were obtained over gridded source data values from the State Soil Geographic database (STATSGO) (Schwarz and Alexander, 1995), which were depth-averaged over all soil layers (Wolock, 1997).

**Table 3.** Soil property predictors.

| Soil property | Unit | Statistics[a] |
|---|---|---|
| Calcium carbonate equivalent | % | Lower/higher bounds |
| Cation exchange capacity | cmolc $kg^{-1}$ | Lower/higher bounds |
| Depth to the seasonally high water table | m | Average and Lower/higher bounds |
| Soil thickness | m | Lower/higher bounds |
| Hydrologic soil group classification | % | Average |
| Soil erodibility factor | dimensionless | Average |
| Permeability | m $h^{-1}$ | Average and Lower/higher bounds |
| Available water content | fraction | Average and Lower/higher bounds |
| Bulk density | g $cm^{-3}$ | Average and Lower/higher bounds |
| Organic matter content | % | Average and Lower/higher bounds |
| Clay soil content | % | Average and Lower/higher bounds |
| Silt soil content | % | Average |
| Sand soil content | % | Average |
| Percent finer than nos.4, 10, and 200 sieve | % | Average and Lower/higher bounds |

[a]: Spatial statistics calculated across the watershed.

## 3.3.3 Geology

The geology predictors used in this study were retrieved from publicly available databases (Wieczorek and LaMotte, 2010c, d) and they can be classified into two subcategories: surficial geology (surface sediment) and bedrock geology. As the predictors, we used fractions of the watershed area corresponding to each of the 45 surficial geology types (Wieczorek and LaMotte, 2010d; Clawges and Price, 1999) and each of the 162 bedrock geology types (Wieczorek and LaMotte, 2010c; Schruben et al., 1994). Details regarding each geology type can be found in Wieczorek and LaMotte (2010c) and Wieczorek and LaMotte (2010d). Note that in geological terminology, rock type or rock composition data are referred to as lithology data. Compared to lithology, structural geology data might be more informative for groundwater studies (e.g., orientation, fracture properties, discontinuity, etc.). However, structural geology information usually requires in-situ investigation, which cannot be expected at ungauged watersheds. Therefore, we consider only lithology data in this study.

 ## 3.4 Data partitioning

This Sect. explains the setup of the holdout method specific to the case study, as well as the partitioning of the predictors into various subsets in order to evaluate the effects of different predictors.

### 3.4.1 Watershed partitioning

Because we cannot evaluate the predictive accuracy at real ungauged watersheds (due to the lack of in-situ data to compare against), we adopt the holdout method to partition the watersheds described in Sect. 3.1 into two mutually exclusive subsets: the training watersheds and the testing watersheds. The testing watersheds will be treated as if they were ungauged, and we only condition the BART models on data from the training watersheds (which are the ex-situ data, with respect to the testing watersheds).

In this study, we define the watersheds in MRB 1 as the testing watersheds and the watersheds in MRB 2 as the training watersheds. The ex-situ data (i.e., data in MRB 2) are used to fit multiple BART models, which are then used to obtain predictive distributions of LNR at all the testing watersheds. There are two reasons for this MRB-based data partitioning:

– For reasons touched on in Sect. 1, we do not consider spatial proximity as a predictor in this study. Separating the two MRBs partly ensures the exclusion of the confounding effect of spatial proximity, and thus the regionalization is solely based on the watershed characteristics.

– Considering the distributions of LNR (Fig. 4(**f**)), the range of values in MRB 2 fully covers the range of values in MRB 1. However, the reverse is not true. It is thus advantageous to train the models with MRB 2 to avoid poor model fitting due to lack of data coverage.

After partitioning the watersheds, we now turn our attention to the partitioning of predictors.

### 3.4.2 Predictor partitioning

As mentioned in Sect. 1, climate variables are among the most important factors in hydrologic similarity at the regional scale, but there might be other controlling factors to consider as well, and the dominance of climate variables may not be always present. To investigate the various effects of different predictors, we conceptually divide the predictors into four sets: (1) climate controls that determine the input amount of water into the system, (2) surface controls that determine the distribution of water at the surface, (3) soil controls that determine the infiltration of water, and (4) lithology controls that indicates the properties of the aquifer. We further break of the first set into three subsets to investigate the effect of dimensionless predictors. Therefore, we define a total of six different predictor sets to build six unique BART models, which are indexed by $k$, $k = 1, 2, ...6$ (Table 4).

Note that the determination of the six predictor sets is guided by a conceptual division of predictors and the idea of testing the relative importance of different categories of predictors under different conditions, instead of aiming for high accuracy and precision. Therefore, by no means is Table 4 an exhaustive list of all possible sets, nor does it necessarily include the best set

15   that leads to the best predictive performance. The design of the six predictor sets simply facilitates the investigation of the effects of various categories of predictors on predictive accuracy and uncertainty.

**Table 4.** Table of the six different predictor sets.

| k | predictors included | Number of predictors |
|---|---|---|
| 1 | $\bar{\phi}$ and $\phi$ | 2 |
| 2 | $\bar{P}$, $P$, and $E_p$ | 3 |
| 3 | All climate predictors: $\bar{P}$, $P$, $E_p$, $\bar{\phi}$ and $\phi$ | 5 |
| 4 | Topography and land cover predictors | 20 |
| 5 | Soil predictors | 48 |
| 6 | Geology predictors | 206 |

### 3.4.3   The benchmark model: without any predictor

In addition to the six BART models, we also build a simple model by using the estimated distribution of LNR at the training watersheds via kernel density estimation (R Core Team, 2018; Sheather and Jones, 1991), without considering any predictor.
20   In other words, this is simply using the distribution of LNR at all the training watersheds as the predictive distribution. This is a model that ignores hydrologic similarity altogether, and it can be considered as an extreme case of the ex-situ prior in Li et al. (2018), with a lot more watersheds and much less stringent criteria of similarity. From this point forward, we refer to this model as the benchmark model, for it is used as a benchmark against which the BART models are compared.

### 3.5   Evaluation of predictive distributions

25   As mentioned in Sect. 2.3, we label each testing watershed by the best-performing model, where the performance is measured based on a metric. Thus, the metric with which we evaluate predictive distributions matters.

In this study, two different accuracy metrics are adopted. The first is the root mean squared error (RMSE), defined as

$$E_{i,k} = \sqrt{\frac{1}{L}\sum_{l=1}^{L}\left(\hat{R}_{i,k}^{(l)} - \tilde{r}_i\right)^2} \qquad (7)$$

where $\tilde{r}_i$ is the LNR data at the $i$th testing watershed, and $E_{i,k}$ is the RMSE of the $k$th model at the $i$th testing watershed. Note that $\hat{R}_{i,k}^{(l)}$ is obtained by following Eq. 6, but now subscripts are added to indicate that we plug in the predictors from the $i$th testing watershed to the $k$th model. This metric evaluates the predictive performance in an estimation problem, where we wish to obtain a "best estimate" of LNR with minimal expected error.

The second metric is the median log predictive probability density (LPD) at the value of LNR observation, defined as

$$L_{i,k} = \text{median}_{l=1,...,L}\left\{\ln\left[p\left(R = \tilde{r}_i|\hat{R}_{i,k}^{(l)}, (\sigma^2)_k^{(l)}\right)\right]\right\} \tag{8}$$

where $L_{i,k}$ is the LPD of the $k$th model at the $i$th testing watershed. The subscript of $(\sigma^2)_k^{(l)}$ indicates the $k$th model. This metric evaluates the predictive performance in a simulation problem, where we wish the realizations from the predictive distributions are likely to be the same as the observation.

In addition to accuracy, we also quantify the predictive uncertainty. This is done by first recognizing the two components of uncertainty for the $k$th model at the $i$th testing watershed:

1. $\sigma_k^2$, which we refer to as the **predictive variance**, and is approximated as the sample median of $(\sigma^2)_k^{(l)}$ over $l = 1, ..., L$, and

2. the posterior variance of $\hat{R}_{i,k}$, which we refer to as the **estimate variance**, and is approximated as the sample variance of $\hat{R}_{i,k}^{(l)}$ over $l = 1, ..., L$.

The predictive variance indicates how informative the inferred predictor-response relationship is, while the estimate variance indicates how uncertain the said relationship is. In this case study we weigh the two components equally, as we wish to obtain an informative relationship with certainty. To that end, we define the **total predictive variance** as the summation of the two components, and use it as the metric of predictive uncertainty in this study.

## 4  Results

As discussed above, we built six BART models (Table 4) with ex-situ data. In-situ predictors were then fed into the models to yield posterior realizations of predictive distributions (Eq. 6). With the metrics of accuracy and uncertainty defined, we are then able to quantify the predictive performance of the BART models, and classify them based on either the RMSE-based labels or the LPD-based labels with the nested tree-based modeling approach. This allows for the investigation of the effects of various predictors under different conditions, which will be presented in this Sect.

### 4.1  Evaluation of predictive distributions

The following subsections present the effects of different predictor sets on predictive accuracy and uncertainty.

#### 4.1.1  Predictive uncertainty

The effect of regionalization with the different predictor sets on predictive uncertainty is shown in Fig. 5. The estimate variance (Fig. 5(**a**)) represents how well the BART models capture the predictor-response relationships. We see that the geology predictors lead to the lowest estimate variance, probably because of the significantly larger number of predictors used (see Table 4). Yet, there is a surprise in Fig. 5(**a**). First, at $k = 1$ and $k = 2$ the estimate variances are generally quite low, despite the low number of predictors. However, at $k = 3$, the estimate variances increase significantly. Intuitively, since aridity is the

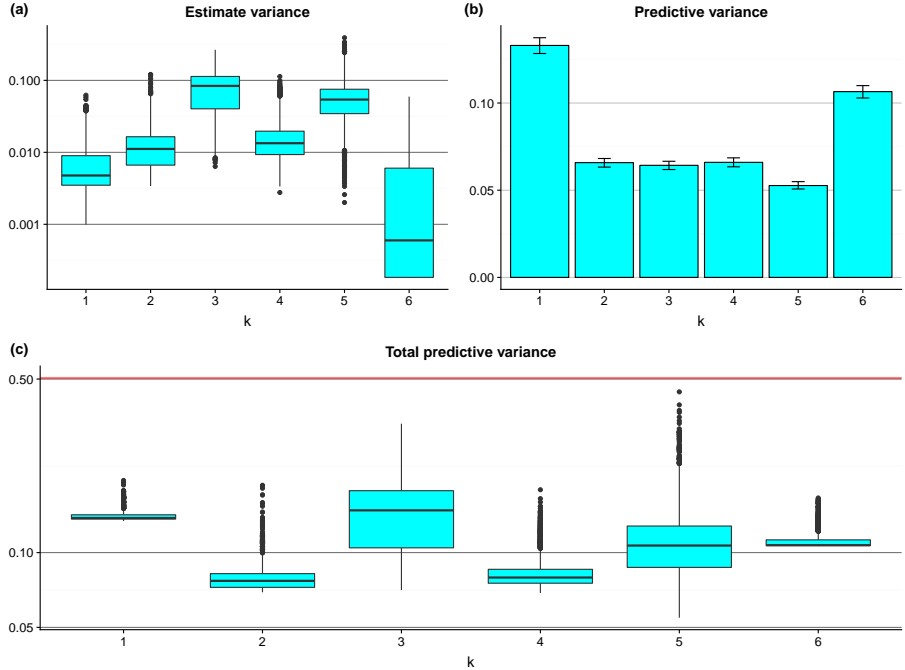

**Figure 5.** The box plots of the estimate variances at the testing watersheds **(a)**, the bar plot of the predictive variances with 95% intervals shown by the error bars **(b)**, and the box plots of the total predictive variances at the testing watersheds **(c)**. The red line indicates the variance of the benchmark model for comparison.

ratio of evapotranspiration to precipitation, one would expect that the variances at $k = 3$ would be similar to, if not lower than, those at $k = 1$ and $k = 2$. One plausible explanation here is that although aridity indices and precipitation/evapotranspiration carry ample information to be extracted and conditioned upon, the respective predictor-response relationships we get might be

25 significantly different. When used together, the BART models were not able to formulate a universal relationship. This will be revisited in Sect. 5.3.

The predictive variance (Fig. 5**(b)**) represents how informative the predictor-response relationships are, which is a different aspect of uncertainty compared to the estimate variance. One could obtain a predictor-response relationship fairly confidently (low estimate variance), but the relationship is less informative (high predictive variance), like that found at $k = 6$. The opposite

30 case is that one could not confidently obtain a predictor-response relationship, but once that relationship is obtained it is quite informative, like that found at $k = 5$.

The total predictive variance (Fig. 5**(c)**) provides an overall metric that considers the above two sources of uncertainties. While the medians are rather similar, the spread of the box plots does vary significantly with $k$. The condensed box plots (e.g., $k = 1$ and $k = 6$) indicate that the total predictive variances are essentially constant throughout all testing watersheds, while the spread-out box plots (e.g., $k = 5$) indicate that the effect of the predictors may vary significantly from one testing watershed to another. This indicates that there might not be one single predictor set that always leads to the lowest uncertainty, and thus

the effects of predictors on predictive uncertainty may vary from one condition to another. That said, regardless of the testing watersheds and predictor sets, the total predictive variance is always lower than the variance of the benchmark model, which clearly shows that regionalization using watershed characteristics definitely improves predictive precision.

### 4.1.2 Predictive accuracy

The effect of regionalization with the different predictor sets on RMSE is shown in Fig. 6. The RMSE of the benchmark
model (Fig. 6**(a)**) at each testing watershed is simply the difference between the sample mean of the ex-situ LNR data and the in-situ LNR observation. For the BART models (Fig. 6**(b)**), it is calculated by the root of the average squared errors over post-convergence MCMC simulations.

     Regardless of $k$, we see that, compared with the benchmark model, RMSE is reduced at least at half of the testing watersheds. Surprisingly, the largest overall RMSE reduction is observed when only the aridity indices are used for regionalization,
indicating that at most of the watersheds tested in this study, aridity similarity implies LNR similarity at regional and annual scales to a high degree. On the other hand, we observe some outliers that have high RMSE reduction at $k = 4$ through $k = 6$, indicating that topography, land cover, soil properties, and geology may not have an overall effect that is as strong, but under certain circumstances, they could still be important factors.

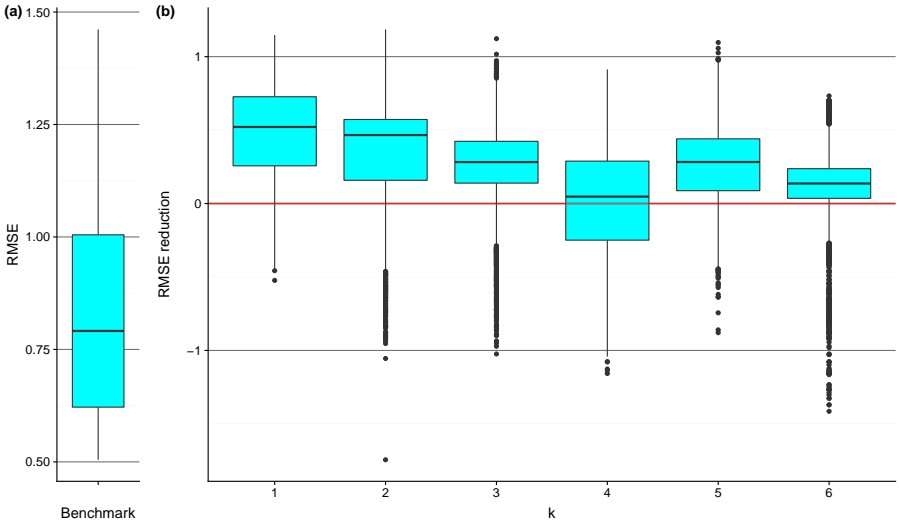

**Figure 6.** The box plot of the RMSE of the benchmark model at the testing watersheds **(a)**, and the box plots of the RMSE reduction introduced by applying the BART models at the testing watersheds **(b)**. The red line indicates zero RMSE reduction for comparison.

     The effect of regionalization with different predictor sets on LPD is shown in Fig. 7. It is immediately clear that the accuracy
improvement is not as prominent as that in Fig. 6. Only when $k = 1$ is LPD increased at most of the watersheds . We also find that all of the distributions of LPD are heavily negatively skewed with a lot of outliers.

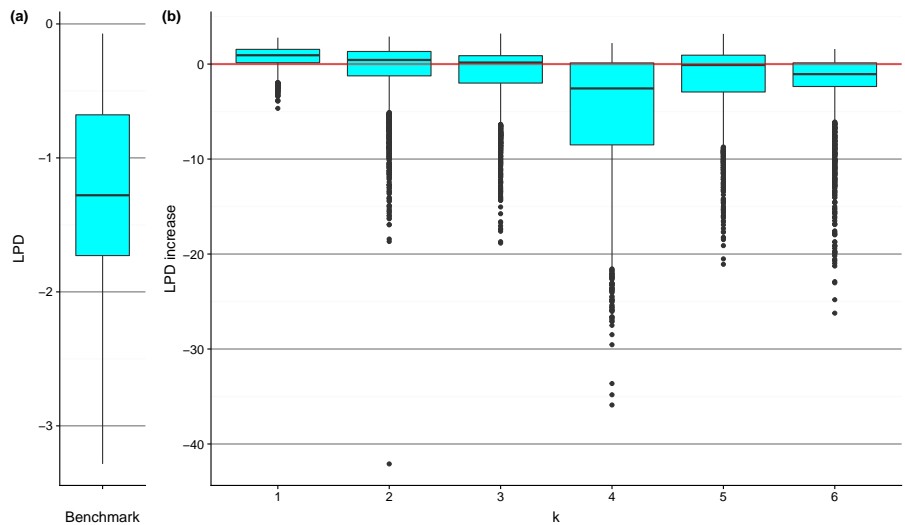

**Figure 7.** The box plot of the LPD of the benchmark model at the testing watersheds **(a)**, and the box plots of the LPD increase introduced by applying the BART models at the testing watersheds **(b)**. The red line indicates zero LPD increase, used for comparison.

Looking at Figs. 5 through 7 together, one can observe the different effects of the predictor sets on predictive accuracy, stemming from the different natures of an estimation and a simulation problem. From the point of view of the overall effect, for $k = 2$ through $k = 5$ (i.e., the predictors other than aridity indices) RMSE is reduced at more than half of the testing watersheds,
but LPD does not increase to the same extent. This suggests that the predictive distributions are centered closer to the in-situ observations due to regionalization, but that the conditioning also significantly reduces the predictive variances, causing the predictive distribution to be too narrow. Therefore, compared to a relatively flat, spread-out, and uninformative or weakly informative distribution, the predictive density decays too quickly when deviating from the predictive mean, resulting in low LPD. This might be a sign of over-conditioning, or the disproportional reduction of predictive uncertainty, as exemplified in
Fig. 8. The cyan curve is an example of an over-conditioned distribution. Although its mean is somewhat close to the true value, the small variance causes rapid decay of probability density; therefore, at the true value (red vertical line) the predictive density is no better than that of the weakly informative or uninformative distributions. How could this ever happen? Take $k = 5$ in Fig. 5 as an example: the predictive variance is small, meaning that the predictive distribution should be rather peaked (just like the cyan curve in Fig. 8). The only way one can get a high predictive density is then to make the predictive mean close to
the true value. Nonetheless, this would be very difficult at some of the watersheds where the estimate variance is large. The only predictor set that improves both RMSE and LPD at most of the testing watersheds is $k = 1$, the aridity indices, and one could expect the corresponding predictive distributions to be somewhat similar to the case of the ideal dark blue curve in Fig. 8.

Over-conditioning can occur when model fitting or model calibration leads to constrained parameters that are, in fact, subject to different forms of model uncertainty (Hutton et al., 2014; Beven et al., 2008), which is an indication of why the determination

of $p(B_k)$ is important. In this case study, we focused more on the variation of $p(B_k)$ under various conditions (to be shown shortly), and less on improving the estimates. However, in another application where the estimates are to be improved, model structure uncertainty should be and can be considered in order to refine the estimates (e.g., via Bayesian model averaging).

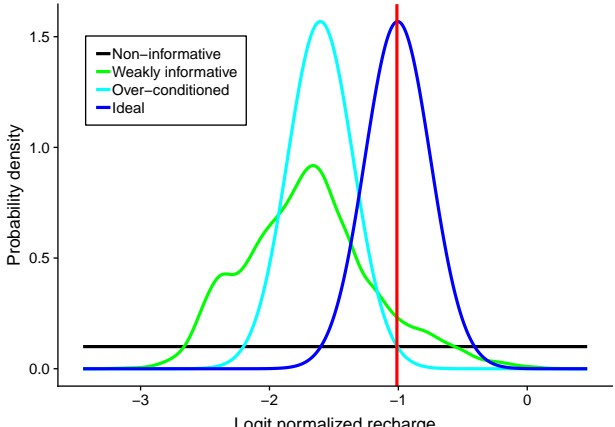

**Figure 8.** An example of over-conditioning: the probability density at the true value (indicated by the red vertical line) of the over-conditioned distribution is not higher than that of the non-informative distribution or that of the weakly informative distribution, not because the conditioning does not work, but because of the disproportional reduction of the variance of the distribution.

## 4.2 Regionalization similarity

The box plots in Fig. 5 through 7 showed different distributions of the predictive performance metrics for the different predictor sets. An interesting follow-up question here would be how model performance varies with watershed characteristics. It was shown that, consistent with previous studies, aridity is indeed the most important controlling factor at regional and annual scales on average, but there are few cases where this aridity dominance is replaced. In other words, how might we identify the conditions under which a specific predictor set could be more informative than others?

To investigate this further, we give each testing watershed two labels: the model with the lowest RMSE, and the model with the highest LPD; we refer to these labels as the RMSE labels and the LPD labels, respectively. The possible values of each label include $k = 1$ through $k = 6$ and $benchmark$, representing the six BART models and the benchmark model, respectively. Then, using all the available predictors, we built two CART models to classify watersheds based on the RMSE labels (Fig. 9), and the LPD labels (Fig. 10).

### 4.2.1 Nesting by RMSE

Fig. 9 shows the variation of the top two best performing BART models and the corresponding $p(B_k)$ values under various conditions, where the performance of each BART model is defined by the RMSE. This variation indicates the regionalization similarity in the study area. At first glance, the available water content (AWC) stand out to be the first indicator of regionaliza-

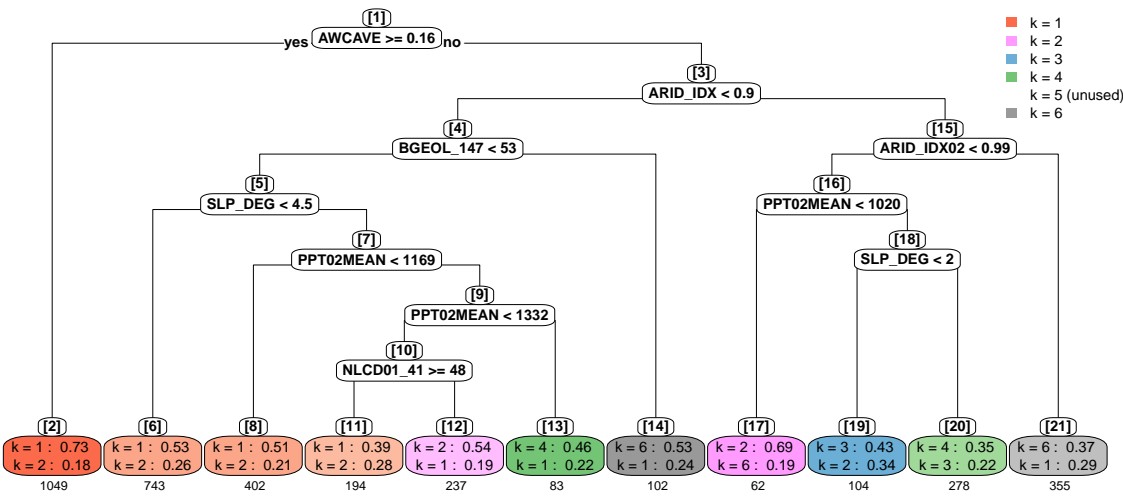

**Figure 9.** CART model classifying the RMSE labels of the testing watersheds. Splitting rules are shown in white nodes, while leaf nodes are colored based on the classification results. For each leaf node, the brightness of the coded color indicates the node impurity (the brighter the more impure), where impurity is defined as the probability that two randomly chosen watersheds within the node have different labels. On top of every node, in brackets, is the node number, provided for convenient referencing. The predictors in the splitting rules are expressed in code names for convenience; a reference list is found in Table 5. For each leaf node, the model of the highest multinomial probability of having the best performance is shown first, which also determines the classification result, followed by the model of the second highest probability, also to indicate the impurity. Underneath each leaf node box is the number of watersheds belonging to the leaf. Note that the legend does not include benchmark because the benchmark model is never the best-performing model at any testing watershed. $k = 5$ is marked as "unused" in the legend because there is no leaf node where $p(B_5)$ is the highest.

**Table 5.** Reference list of the splitting variables in Fig. 9 and Fig. 10.

| Code name | Splitting variable |
|---|---|
| AWCAVE | Average available water content |
| ARID_IDX | Long term average aridity index |
| ARID_IDX02 | Aridity index in 2002 |
| PPT02MEAN | Annaul precipitation in 2002 |
| BGEOL_147 | % area of Paragneiss and Schist bedrock |
| SLP_DEG | Average slope in degree |
| NLCD01_41 | % area of Deciduous Forest |

tion similarity (Fig. 9 node 1): at watersheds with high AWC, aridity stands out as the dominant factor, which is consistent with the previous studies cited in Sect. 1. However, there is a potential risk if one uses aridity as the primary indicator of hydrologic

similarity regardless of AWC. In previous studies, AWC was found to be an important predictor correlated with surface runoff, baseflow, and groundwater recharge (Arnold et al., 2000), and it was among the most important parameters to which water balance models are sensitive (Finch, 1998). In the current study, we are not claiming that AWC cannot be a predictor, but

rather, we are suggesting a hierarchical structure in which AWC is placed —together with other predictors —to help estimate LNR at ungauged watersheds. Since AWC is governed by field capacity and wilting point, it is an indicator of the storage capacity of the soil for usable/consumable water: the larger the storage capacity, the higher the degree to which the system is supply-limited, thus pointing to aridity. If the storage capacity is low, on the other hand, the more complicated interplay among various predictors needs to be considered, and one cannot simply assume that aridity is the primary indicator of hydrologic

similarity. We also found the soil organic matter content a quite competitive surrogate for AWC, meaning that if organic matter content was used here instead of AWC, we would end up with a slightly less accurate but overall similar classification. We conjecture that this is because of the high positive correlations between organic matter content and AWC (Hudson, 1994).

Further down the classification tree, watersheds with lower AWC are classified roughly as arid or humid watersheds by the long-term aridity index. For the more humid watersheds (Fig. 9, nodes 4 through 14), regionalization similarity is controlled

by different predictors, but the dominant predictors for LNR estimation are almost always the climate variables (nodes 6, 8, 11 and 12, which contain 1576 watersheds in total). Only at a handful of watersheds (nodes 13 and 14, which contain only 185 watersheds in total) are aridity indices not dominant. However, some interesting conjectures can be made by taking a closer look at these two nodes.

Node 14 is a small but unique cluster, featuring watersheds that have low AWC, are humid, and have relatively homogeneous

paragneiss and/or schist bedrock. Both of these bedrock types belong to the category of crystalline rock, and often feature layering in a particular orientation. The groundwater movement in such rock formation often depends on foliation, i.e., rock breaks along approximately parallel surfaces, which affect the direction of the regional groundwater flow (Singhal and Gupta, 2010). Hence we observe a condition where the ample water supply cannot be substantially held by the soil due to low AWC, and the regional groundwater movement might be controlled by bedrock layering and foliation. Low AWC is an indication

of less clayey soils, and implies that infiltration/percolation through the soil layer might be facilitated by relatively higher permeability. Water could thus easily enter the bedrock layer, which is rather horizontally homogeneous. To that end, those predictor sets other than $k = 6$ become less informative, while the predictor set $k = 6$ becomes relatively more informative. In fact, these watersheds are mostly the positive outliers at $k = 6$ in Fig. 6(**b**), where the predictive power of the geology predictors is at its best.

Node 13 features watersheds that have low AWC, are humid, are not dominated by homogeneous paragneiss and/or schist, have a relatively steep average slope, and have a large amount of annual precipitation. The low aridity is primarily driven by precipitation rather than evapotranspiration. In fact, these watersheds are mostly outliers featuring extremely low aridity index (below 0.65) due to ample precipitation. Under such condition, evapotranspiration is expected to operate to its full potential, i.e., it is shifting from water-limited state to energy-limited and canopy-controlled state. In addition, as evapotranspiration is near its full potential, the drainage of the excess precipitation would be controlled by the topography of the watershed (e.g., the slope and the sinuosity of the stream). Fast drainage leaves less water available for infiltration and recharge, and vice versa.

5    To that end, the land cover type and topography now start to play a dominant role in hydrologic similarity. It is noteworthy to point out node 20 here. Node 20 features watersheds that are relatively humid among the arid watersheds ($\bar{\phi}$ in the range from 0.9 to 0.99) and have ample precipitation. The similarity of node 20 with node 13 supports our conjecture that the dominance of land cover and topography predictors is due to the precipitation-driven humid environment that is relatively more capable of catering to the evapotranspiration water demand and features excess precipitation.

10    On the other side of the tree (Fig. 9, node 15 through 21), the resulting classification is quite diverse, and the impurity of each node is relatively high. Aridity no longer plays the dominant role, and the hierarchical similarity structure becomes complicated that it is difficult to make straightforward physical interpretations. The most important message we get is the significant risk one would face if one considers aridity, or any climate variable in general, as the primary indicator of hydrologic similarity when AWC is low and aridity index is high. In summary, although climate predictors are still the most important ones on average, within the context of the hierarchical similarity we have identified certain conditions under which either non-climate predictors become dominant or no dominant predictor set can be straightforwardly identified, all of which contribute to the

understanding of the dynamic hydrologic similarity.

### 4.2.2   Nesting by LPD

The classification of the LPD labels is shown in Fig. 10. In general, the root part of the classification tree (node 1 through 3) is quite similar to that found in Fig. 9, where AWC and long-term aridity define two sequential overarching separations of watersheds. However, further down the tree the leaf part is significantly different. The classification essentially leads to only

three big clusters (Fig. 10, nodes 2, 7, and 9), and the other leaf nodes only contain a few watersheds. Node 9 features arid watersheds with low AWC, where we end up with a highly impure leaf node, and even the highest multinomial probability is only 0.27. No further splitting rule could significantly reduce classification error. This is supportive towards our previous argument that when aridity index is high and AWC is low, it is risky to resort to climate variables for hydrologic similarity, as shown here that it is difficult to even identify a dominant predictor set. As mentioned in Sect. 4.1.2, underestimation of the

predictive variance ($\sigma_k^2$) leads to low LPD, and thus it is difficult to make physical interpretation our of the results in Fig. 10, except for node 1 through 3, which are quite similar to their counterparts in Fig. 9. Therefore, with the LPD labels we are only able to identify the overarching regionalization similarity controlled by AWC and long-term aridity.

    RMSE and LPD represent views of predictive accuracy in an estimation problem and a simulation problem, respectively. Intuitively, if one only considers unimodal predictive distribution with limited skewness, a high predictive density at a value directly implies a closeness of the distribution central tendency to that value. However, the reverse is not necessarily true:

either over- or underestimation of variance might possibly lead to low predictive density, even if the mean is close to the target value (e.g., Fig. 8). Based on whether RMSE or LPD is used as the accuracy metric —which implies the scope of LNR estimation —we can observe some common features as well as some distinctions of the structure of the hypothesized hierarchical similarity.

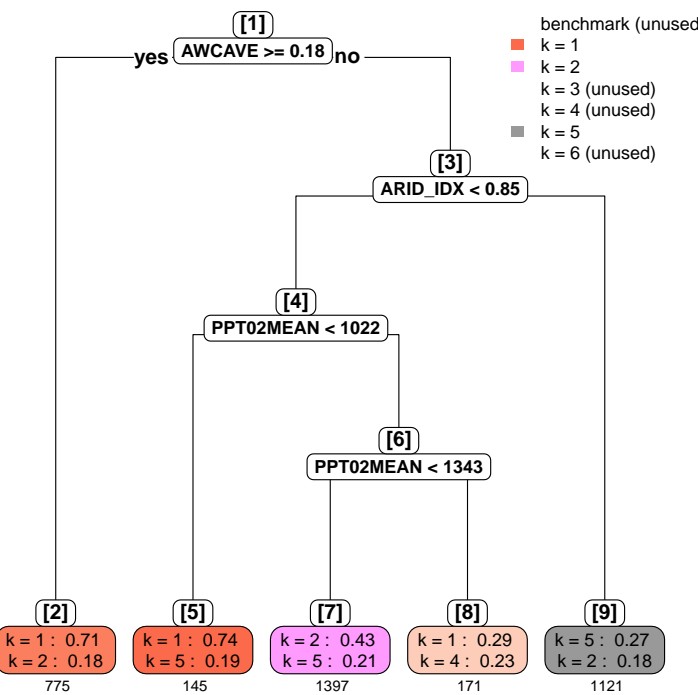

**Figure 10.** Same as Fig. 9, except that here the classification is done using the LPD labels. The predictors in the splitting rules are expressed in code names for convenience; a reference list is found in Table 5.

Fortunately, regardless of the metric of predictive accuracy, in both Figs. 9 and 10 the first three nodes are remarkably consistent, and the effect of the metric of predictive accuracy is only manifested at watersheds with low AWC. This supports the suggestion that AWC plays a pivotal role in hydrologic similarity for mean annual LNR estimation.

## 5   Discussion

In this section, we revisit the two research objectives pointed out in Sect. 1 by discussing the key features of the approach, the key findings from the case study, as well as the limitations of the case study.

### 5.1   The nested tree-based modeling approach

The nested tree-based modeling approach proposed in this study is essentially a coupling of BART and CART. As demonstrated in Sect. 2, both BART and CART are independent of the physical background, and are pure data-driven machine learning techniques. Therefore, in principle as long as there are data, the nested tree-based modeling approach is applicable like any other data-driven approach. However, one may argue that (1) the in-principle applicability does not set the nested tree-based

modeling approach apart from other data-driven machine-learning approaches, and that (2) it would be counter-intuitive to advocate a data-driven approach with a seemingly data-rich case study (here "data-rich" refers to the fact that each MRB consists of thousands of watersheds, see Sect. 3.1) when the study actually emphasizes ungauged watersheds.

Our explanation starts with explaining two significant advantages of the nested tree-based modeling approach. First of all, the greatest advantage of BART (as mentioned in Sect. 2.2) is that it outputs the posteriors of the model parameters, which could lead to posteriors of the target response. The advantage of having the posteriors is that the users/modelers can then derive the desired information at will, such as percentiles, moments, information gain, or the posterior mean and variances like what was demonstrated in the case study. Conditional simulation is also made easy when the posteriors are available, opening the door for Monte-Carlo analyses. Second, following the statement that one can obtain the statistics or representative metric of interest, the nesting of BART models under CART can be done with the said metric, resulting in the corresponding probability mass function of the plausible BART models. For example, the classification shown in Fig. 9 is based on RMSE, which is then based on the posterior mean values. This is essentially a proposal-comparison-based consideration of model structure uncertainty.

How do the aforementioned two advantages of the nested tree-based modeling approach justify the use at ungauged watersheds? First, of course the performance of the model depends on the quality and the quantity of training data. In this sense all modeling approaches are the same, and applying BART does not disproportionally enhance the predictive accuracy when the data are limited. However, what sets BART apart is the Bayesian feature that accounts for model parameter uncertainty properly in the form of conditional distribution, which cannot be done as easily with only a few point estimates or a few posterior statistics. Second, uncertainty exists not only for the model parameters but also for the models themselves. The nested tree-based modeling approach can help us obtain an informed empirical probability mass function, $p(B_k)$, of the plausible BART models (which was also exemplified in the case study). The fact that at ungauged watersheds in-situ data are absent and ex-situ data can be limited in quantity and/or quality accentuates the importance of uncertainty quantification, and the nested tree-based modeling approach offers a Bayesian solution to that, making itself not only applicable but also advantageous at ungauged watersheds.

One may then argue that how would a modeler make an informed proposal of plausible BART models in the first place? This is where physical knowledge come into play, and the proposal is indeed case specific. This is why we proposed the hypothesis of hierarchical similarity, which can be integrated with the nested tree-based modeling approach to study the behavior of a dynamic hydrologic similarity system, like what was demonstrated with the case study. Unlike the generality and the merits of the nested tree-based modeling approach, our findings regarding the variation of $p(B_k)$ and the shifts in dominant controlling factors of recharge are indeed specific to the context of the case study, which will be discussed next.

## 5.2   The hierarchical similarity hypothesis and the shift in dominant physical processes

With BART's ability to simultaneously model non-linear and/or interaction effects and present uncertainty in a fully Bayesian fashion, we are able to show how the controlling factors of hydrologic similarity vary among different watersheds, among

different conditions, and among different accuracy metrics. These are all manifested in the case study under the context of the hierarchical similarity hypothesis.

Climate variables have been identified as the dominant factors in previous studies (see Sect. 1), and they are indeed on average the most dominant factors in our case study. However, the hierarchical similarity shows potential risks if one resorts to climate variables to define hydrologic similarity without considering other physical watershed characteristics, especially the soil available water content.

The details of the hierarchical similarity are inferred from the data in the fashion of supervised machine learning, using six BART models and one benchmark model nested under one classification tree. It is of great importance to have two levels in such a system, as it allows for identification of the shifts of dominant factors under different conditions. These shifts indicate shifts in dominant physical processes, as exemplified by node 13 and 20 in Fig. 9 where we observed the shift from water-limited evapotranspiration to energy-limited evapotranspiration. Therefore, we conjecture that it is the shift in dominant physical processes that is driving, and thus is reflecting, the shift in the controlling factors of hydrologic similarity under different conditions.

## 5.3 Limitations of the case study

Here, we provide discussions about the limitations of the case study from the aspects of the data set, the target response, and the partitioning of data.

### 5.3.1 The scale of the target response

A major limitation of the case study is that the target hydrologic response is the logit normalized watershed-averaged annual groundwater recharge. This is a large-scale spatiotemporally homogenized response, and in this study, the data were based on baseflow analyses. To that end, a working assumption about the reliability of the baseflow analysis was made without rigorous proof (see Sect. 3.1). The findings of the case study are all under the context of this working assumption, and thus, they should not be applied to recharge/LNR at other spatiotemporal scales or to other hydrologic responses without careful considerations.

### 5.3.2 The MRB-based partitioning of watersheds

Although we tried to justify the MRB-based partitioning by the reasons listed in Sect. 3.4.1, we acknowledge that this may not be the best partitioning method for demonstrating the full potential of the estimating power of BART. An associated limitation is identified, which stems from the data not covering a desirable range of values. An example was already presented in Sect. 3.4.1 and Fig. 4. As discussed in Sect. 5.1, the limitations in the data accentuate the advantage of our approach regarding the consideration of uncertainty, but it is also recognized that it could be challenging to discover the same findings if MRB 1 provided the training data for MRB 2, which is part of the reason why we kept the MRB-based partitioning.

Another case of lack of data coverage can be found in our climate predictors data. Since aridity index is the ratio of potential evapotranspiration to precipitation ($\phi = E_p/P$), one might be surprised by the differences among the cases of $k = 1$, $k = 2$,

5    and $k = 3$ in the results. The main reason is revealed in Fig. 11. The $E_p$ values at the training and testing watersheds are so distinct that, essentially, all the testing watersheds are outliers from the point of view of a BART model trained at the training watersheds. On the other hand, the $\phi$ values at the training and testing watersheds share the range from about 0.6 to 1.2, and only differ at the two extreme ends. In other words, the predictor-response relationships inferred by using $\phi$ can be transferred due to the overlapping range (Fig. 11(**c**)), but the relationships inferred using $E_p > 1000mm$ cannot be effectively transferred

10   to watersheds with $E_p < 1000mm$ (Fig. 11(**b**)). Although it is not shown, a similar case can be found by comparing $\bar{\phi}$ with $E_p$.

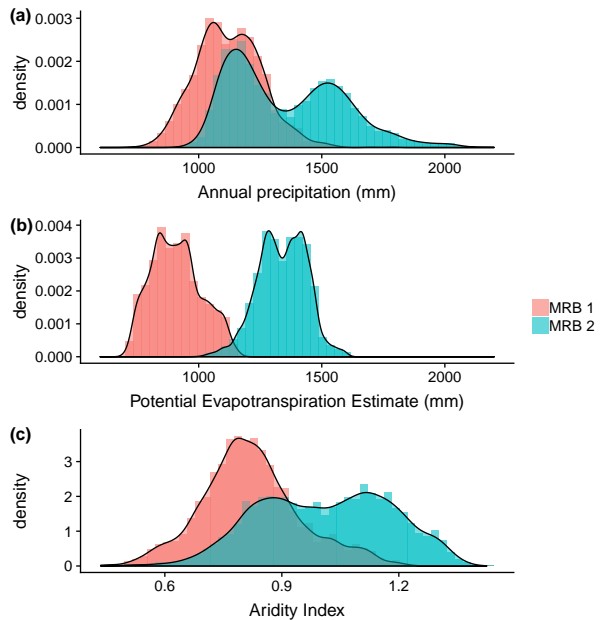

**Figure 11.** Distributions of (**a**) $P$, (**b**) $E_p$, and (**c**) $\phi$, at watersheds in MRB 1 (the testing watersheds) and MRB 2 (the training watersheds).

Although this might have been avoidable by using a more sophisticated design of cross-validation, we kept the MRB-based holdout method on purpose. In addition to the reasons that were explained in Sect. 3.4.1, another motivation is that, in reality, the data at hand come in as is. This means there is no guarantee that the measurements will cover a particular range or that the watershed characteristics of the ungauged watersheds of interest are within a desirable range. The prevailing superiority of $\phi$ and $\bar{\phi}$ over $P$, $\bar{P}$, and $E_p$ found in our results shows an important advantage of dimensionless predictors, that they tend to be more transferable from one site to another, and hence, they may be more suitable for studies targeting ungauged watersheds.

### 5.3.3   Limited temporal data coverage

Another limitation is the lack of temporal coverage. Given limited data coverage along the time axis, in the case study we only studied the LNR in the year of 2002, and we considered two types of climate predictors: those from the same year and those from the long term average. However, being the recharge process highly non-linear, it is not impossible that some predictors

representing the antecedent conditions, such as precipitation from years prior to the year of 2002, could affect the LNR in the year of 2002. Not having multiple years of climate data prevents us from testing the effects of antecedent conditions or the

effects that take place at various multi-year scales, and thus it is clearly a limitation of the case study. Because of this limitation, we made a steady state working assumption (mentioned in Sect. 3.1), with which we assume that the effect of climate predictors from the previous years are captured by the long term average predictors, and also assume negligible effect of climate change. While acknowledging the inclusion of multiple years of climate data could have made an impact, note that the highly consistent roots of the trees in Figs. 9 and 10 are based on soil AWC and the long term average aridity index, both of which are expected

to be relatively insensitive to the inter-annual variation of climate predictors. Therefore, we expect the findings corresponding to the roots of the trees in Figs. 9 and 10 to be relatively less affected by the limitation of not having multiple years of climate data.

### 5.3.4   Non-comprehensive list of plausible models

The proposal of plausible BART models was guided by a conceptual understanding and grouping of the available predictors.

Like mentioned in Sect. 3.4.2, our proposal does not cover a comprehensive list of plausible models, nor does it necessarily include the "best" or the "true" model. The effect of different proposals of plausible BART models, which represents different perspectives of the conceptual understanding of the underlying physics, was not investigated in the case study, and remains as an interesting follow-up that could be pursued in future studies.

## 6   Conclusions

In this work, we proposed a nested tree-based modeling approach with three key features: (1) full Bayesian quantification of parameter uncertainty, (2) non-linear regression in order to model the predictor-response relationship, and (3) proposal-comparison-based consideration of model structure uncertainty. We applied the nested tree-based modeling approach to obtain logit normalized recharge estimates conditioned on ex-situ data at ungauged watersheds in a case study in the eastern U.S. We hypothesized a hierarchical similarity to explain the variation of the probability mass function of plausible models, and thus to

investigate the behavior of a dynamic hydrologic similarity system.

The findings of this study contribute to the understanding of the physical principles governing robust regionalization among watersheds. Firstly, consistent with previous studies, we found that the climate variables are on average the most important controlling factors of hydrologic similarity at regional and annual scales, which means a climate-based regionalization technique is on average more likely to result in better estimates. However, with our hierarchical similarity hypothesis we revealed certain conditions under which non-climate variables become more dominant than climate variables. In particular, we demonstrated how soil available water content stood out to be the pivotal indicator of the variable importance of aridity in hydrologic similarity. Moreover, we showed that with hierarchical similarity one could identify shifts in dominant physical processes that are

reflecting shifts in the controlling factors of hydrologic similarity under different conditions, such as water-limited evapotranspiration versus energy-limited evapotranspiration, or homogeneous and foliated bedrock versus heterogeneous bedrock. As the

controlling factors change from one condition to another, the suitable regionalization technique also changes. We demonstrated how the hierarchical similarity hypothesis could indicate mechanisms by which available water content, aridity, and other watershed characteristics dynamically affect hydrologic similarity. The nested tree-based modeling approach can be applied to identify plausible sets of watershed characteristics to be considered in the regionalization process.

The contributions of this study may be viewed differently depending on individual cases. In a situation where groundwater recharge is the ultimate target variable at ungauged watersheds, the nested tree-based modeling approach offers a systematic way to obtain informative predictive distributions that are conditioned on ex-situ data. In a difference case, where recharge estimation at ungauged watersheds is but one component of a greater project, the aforementioned informative predictive distributions can be treated as informative ex-situ priors, which could be further updated and/or integrated into simulation-based stochastic analyses where recharge is an input/component of other models/functions. At ungauged watersheds that will become gauged in the foreseeable future, the informative predictive distributions again serve as informative ex-situ priors that could guide the design of the sampling campaign, as different recharge flux magnitudes require different quantifying techniques (Scanlon et al., 2002; Healy, 2010). The hierarchical similarity hypothesis offers one plausible explanation of the dynamic nature of hydrologic similarity, which affects the application of regionalization. Lastly, it should be pointed out that the nested tree-based modeling approach is independent of the target response and the predictors of interest, so it could be integrated into future studies within or beyond the field of hydrology to study hierarchical predictor-response relationships.

*Competing interests.*   The authors declare that they have no conflict of interest.

*Acknowledgements.*   For this study, Ching-Fu Chang was financially supported by the Jane Lewis Fellowship from the University of California, Berkeley. The authors thank Dr. Sally Thompson and Dr. Chris Paciorek for the inspiration of this study. The authors also appreciate the helpful comments from two anonymous reviewers.

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
