# Peer review of "Regionalization with Hierarchical Hydrologic Similarity and Ex-situ Data for the Estimation of Mean Annual Groundwater Recharge at Ungauged Watersheds"

_Hydrology and Earth System Sciences, 2018_

## Referee Comment (RC1) · Anonymous Referee #1 · 7 Feb 2019

The manuscript by Chang and Rubin "Regionalization with Hierarchical Hydrologic Similarity and Ex-situ Data for the Estimation of Mean Annual Groundwater Recharge at Ungauged Watersheds" submitted to Hydrolog. Earth Syst. Sci. Discuss presents a new method for the estimation of mean annual groundwater recharge at ungauged watersheds based on the concept of hierarchical hydrologic similarity. Such a similarity is performed through a nested tree-based modelling approach, accounting for both the predictor-response relationship via a Bayesian Additive Regression Tree model and the predictor-predictor relationship via a CART model. The manuscript is within the

scope of the Journal and it is certainly of interest for the Readers of HESSD. It is well written, in both presenting the research framework and previous literature and showing and discussing results. However, I have some minor concerns that in my opinion should be addressed before a possible publication on HESS. 1. Sect. 3.2 Climate. The choice of predictors is clearly related to the availability of data. In particular, being available recharge data only for 2002, one of the possible predictors is necessarily the precipitation in 2002 (and consequently the aridity index). However, being the recharge process highly non linear, and having demonstrated a posteriori the importance of the climate factors, I feel that the choice of a given year for precipitation (not the predictor precipitation in a given year) can potentially lead to different results, in terms of both predictive uncertainty and predictive accuracy. In particular, one of the strongest conclusions of the manuscript ("The most important message we get is the significant risk one would face if one considers aridity, or any climate variable in general, as the primary indicator of hydrologic similarity when AWC is low and aridity index is high") could be affected by the selected predictors. I think that this issue should be mentioned and discussed. 2. Section 3.4.1 (watershed partitioning). Line 21 ("Considering the logit normalized ... due to lack of data coverage"). I totally agree with Authors. However, in my opinion this could be an important limitation in the evaluation of the proposed method. Maybe the evaluation of the size of the data set for training, with respect to the size of data set for testing is somehow out of the scope of the manuscript, however this issue should be a little presented and discussed at least in the "limitations of the case study" sect. 3. Sect.5.2.1 ("Scale of the target response"). Here the Authors refer to Healy (2010) to face the problem of the scale of the target response, explicitly referring to the limitation of a baseflow analysis. They state: "At an ungauged watershed, it is unlikely that one would have enough data to verify the answers to these three questions". I agree, but the same comment can be done for predictors: "at an ungauged watershed, it is unlikely that one would have enough data for predictors". I know, this is a methodological paper, but methodology is addressed to a very practical problem. The issue of the transferability of the proposed method to real cases (which means the

transferability in case of scarce availability of data) should be somehow faced, at least qualitatively. In the adopted case study, you have all possible data except recharge, which is an "extreme case". 4. Sect. 5.2.2 ("artefact due to the partitioning of watersheds"). Comments on the not-transferability of information outside the overlapping range of predictors and/or targets are certainly shareable. It is shown here that one can not infer outside the observed ranges. However, if in an ungauged watershed I'm exploring a known or unknown horizon is indeed . . . unknown Minor comments on figures: 1. Figure 1. As this figure shows the BART approach in a general way, I would avoid referring to the specific case (annual recharge estimation) and adopt a more general notation in the panel (c) (in the present version of the manuscript "ex-situ" predictors and "ex-situ recharge data") 2. Figure 2. Figure on the left side is illegible. If it does not contain useful information, please cancel boxes. 3. Figure 3. I think it could be useful adding also the distribution of long average P and long average Ep 4. Figures 8 and 9. Node numbers are illegible. As Authors often comment results on one given node, increasing the font size could be useful.

―――――――――――――――

---

## Referee Comment (RC2) · Anonymous Referee #2 · 9 Feb 2019

Summary "Regionalization with Hierarchical Hydrologic Similarity and Ex-situ Data for the Estimation of Mean Annual Groundwater Recharge at Ungauged Watersheds" by Chang and Rubin, outlines a new approach to predict groundwater recharge in ungauged watersheds and assess process controls. The approach used is based upon hierarchical hydrologic similarity and uses a nested tree framework. Bayesian Additive Regression Tree models were developed and evaluated. Following this, each of the BART model selection was assessed using Classification And Regression Trees. I believe this manuscript is within the scope of the journal.

[Figure]

I would accept this paper with major revisions to be made. Firstly, I think some time should be spent making this manuscript better organised and easier to follow, especially figures and abbreviations. The General Notes section below highlights a few improvements to make. Secondly, one of the goals of the study was to develop a method to predict recharge in ungauged watersheds. However, it was only tested in a data abundant scenario using modelled baseflow data and therefore may not be practical when faced with more limited observational data. Point 3 of the main suggestions discusses this further and point 4D suggests a potential way forward. However, as a method to help understand which controls for groundwater recharge are being reflected in the recharge estimation outputs, I think the approach you have used is very good. It helps the reader understand the interaction between different controls and could be used in other hydrological studies or research areas. If the authors cannot address the issue concerning whether the approach is applicable in predicting recharge in ungauged watersheds using solely observational data, perhaps focussing on process understanding could be a way forward.

General Notes

1. Organise objectives more clearly.

Currently I am finding objectives in different parts of the paper which sometimes sound only partially similar. Two are at the end of the introduction (where they should probably go) "Estimate mean annual groundwater recharge at ungauged catchments" and "Propose new approach to understand hydrologic similarity with regards to mean annual groundwater recharge". Others are found at Page 7, lines 12-14 ("To understand what behaviour we can expect in ungauged areas") and page 8 line 6 ("To reveal controlling factors of dynamic hydrologic similarity system for mean annual groundwater recharge"). Put all objectives at the end of the introduction section.

2. Organise sections/paragraphs more clearly.

Sections 3.4 to 3.5 are very methodological, describe the method used in testing ungauged areas and how to evaluate the models and should be moved to the method section. These sections don't seem to give further details about the case study and should be moved to the method section.

3. Organise sections/paragraphs more clearly.

Page 8, Lines 20-30 (Section 2.4). Explains the data partitioning, but data partitioning is then explained again in section 3.4. Integrate this explanation with section 3.4.

4. Organise sections/paragraphs more clearly.

Page 9 Line 5:" Section 2.3 is not demonstrated in this case study"; Remove section 2.3 as it is not used and therefore is not providing information which is useful for this study.

5. Organise sections/paragraphs more clearly.

Page 10, Lines 20-28: you start discussing transformations made to recharge/baseflow data in a section which is supposed to be about climate variables. Move this to section 3.1.

6. Method section has too many equations.

Instead of relying upon equations, explain your methodology in laymans terms (simply) and then reference the papers which give the more explicit details about the method. The existing text describing the methodology should also be simplified.

7. Describe CART in method section.

CART is used to evaluate the different BART models, but no explanation of CART is made in the method section and some readers may not be familiar with this method. Describing CART may also be a useful step in explaining BART which seems to be a more complex version of CART. A schematic of how the BART models fit within the CART framework should also be provided.

8. Explain the terms ex-situ and in-situ data.

These terms come up quite a lot and I believe describe how the data is partitioned for training and testing. However, the terms are used prior to the section which describes data partitioning. Please provide definitions for these terms in the abstract or introduction where they are first used. This is particularly important as readers could get confused with thinking in-situ means from in-situ observations such as water table fluctuations or tracers, which I don't think is what you mean by the term.

9. Provide maps comparing the benchmark model to the other BART model estimates. This would make for an interesting comparison, showing the spatial variation from each of the models, which the reader cannot learn form figures 4-6.

10. Improve the naming conventions for the predictors and the models.

Some of the predictor names are not intuitive at all i.e. NLCD01_41, therefore when it is shown in a figure, I can't learn anything from this. Additionally, instead of calling the BART models 1-6, call them aridity, P&PET, all climate, Soils, Geology, topo. Then I can understand what each model in the figure represents without having to flip to the tables at the back.

11. Legends explaining colour codes on figures are needed. Colours being used in figures 8 and 9 need explaining in a legend.

Main Suggestions

13. Are you assessing recharge ratio or annual recharge?

Despite often referring to annual recharge in the text as the signature you are evaluating, Page 10 line 22 states that annual recharge was normalized by precipitation. This therefore means you are analysing the recharge ratio, a different signature to annual recharge, which instead quantifies the proportion of precipitation being converted to recharge. I do think the recharge ratio is a good signature to use to evaluate recharge controls beyond precipitation. Therefore, I suggest changing the text to say

you are evaluating the recharge ratio to understand recharge processes and predict the recharge ratio in ungauged watersheds.

14. The method applied here to predict recharge in ungauged watersheds needs to be trialled in a less data abundant scenario, otherwise I don't think you can claim that this is a method which can predict recharge in ungauged watersheds.

You have trialled this method in a data abundant scenario using modelled data (gridded baseflow data) as your test case. Regression tree models typically perform better with increasing observations to learn from and therefore any adequate performance you achieved in this data rich scenario may not be achieved when applied to in-situ observations of recharge.

I suggest that instead of using all the estimations within the two basins for training and testing, sample only a fraction of them and then see how well the models perform. Idea - If you sample only data points (for training and testing) where the streamflow data underlying the baseflow data has been evaluated against observations, this would then give an indication of how well the model works in real-world scenarios. This resultant model could then be used for prediction across the entirety of the two basins. You could then perform a second test on the basin wide modelling results, comparing these to the original gridded baseflow dataset you used. From this you could then learn how well the method trains and tests in a more likely scenario of limited data, as well as see how well the model trained and tested on limited data performs in data abundant scenario.

This would then give a better indication of whether the method can be practically used for prediction in ungauged watersheds.

15. It would be useful to know what number of observations is approximately needed to adequately train and test the BART models for prediction purposes.

16. If you decide not to test your prediction method on a more realistic scenario, where data is not as abundant as modelled estimates, I would suggest focusing on the objec-

tive of learning about process controls at large scales.

17. I like the approach of having separate BART models for different types of environmental characteristics. This provides a very nice way of understanding what are the important controls for recharge in different areas.

18. Topography and Landcover characteristics should have their own models, similarly to geology and soils.

No explanation was given as to why you grouped these two domains together and not the others. I think it would be better to be consistent and have separate BART models for each. You could learn something extra from doing this.

19. Aggregate some of geological predictors into their higher-level lithological groupings.

The number of Geology predictors in comparison to those available for climate, soils etc is too high. This could likely give the Geology BART model an unfair advantage in comparison to the other models. It also means that any BART tree for Geology is unlikely to be very informative as it will be difficult to understand what all the sub lithological categories mean.

20. Don't partition data by which basin it is in.

In the discussion you highlight aridity index was a good model as the data from both basins overlapped with respect to aridity, something which wasn't the case for Mean annual precipitation and evapotranspiration. You argue that this by saying that information can come in ad-hoc and therefore your testing it in what could be a realistic situation. However, I would argue that the modeller always has the opportunity to decide how to partition their training and testing data. Therefore, in order to improve the robustness of the estimations, should be looking to make the distributions of each attribute in the training and testing samples, overlap.

I also think you should be sampling less data for training and testing purposes in line
with point 14.

21. Explain the benchmark a bit further so it is clear what you did.

I think the benchmark was a good way to see how the different models could improve recharge estimates and then learn something about process controls. But I think a bit more explanation of how you determined the benchmark is needed, especially as how we evaluate groundwater systems at large scales is an interesting topic. Would also be good to see how this benchmark then compares to the 6 BART models (not shown in Figure 4 and some maps would be good).

Minor Suggestions

22. Page 4, lines 10-13: do you need to say this?

23. Page 6, Figure 1: Would be good to have a schematic of how the BART models are used within the CART framework.

24. Page 6, Lines 16-17. Don't understand what this means.

25. Page 8, lines 22-23. Does each plausible predictor set i.e. k=1 have multiple BART models? Or does each predictor set have just 1 BART model?

26. Page 8, Lines 9-17: I found the explanation of the two levels of similarity confusing and think it would benefit from rephrasing.

27. page 9 Line 5:" Section 2.3 is not demonstrated in this case study"; Do you need Section 2.3?

28. Page 9, Figure 2: Why not show a map of the baseflow estimates in MRB1 and MRB2. This would show the reader where the two basins are as well as allow them to see the variability of baseflow in the region.

29. Page 10, lines 3-5: I think you may need a reference to show that baseflow analysis is suitable for annual recharge estimation in the eastern US to justify yourself. You

justify it in the discussion (section 5.2.1) but should do it earlier on.

30. Page 10, lines 15-19: If you are already looking at climate variables for the year 2002 which correspond to the baseflow data you are using, why are you also interested in the long term climate variables?

31. Page 10, lines 15-17: For the long-term climate variables it would be better to use data which is consistent with regards to temporal extent (i.e. 1970-1990 for both P and Ep).

32. Page 10, Lines 21-22: If normalizing recharge by precipitation, your analysis is then looking at the recharge ratio rather than annual recharge which is a different signature.

33. Page 10, lines 21-28: Why are you discussing transformations made to your recharge data in a section which is supposed to be about the climate variables you are using? This would be better in the recharge estimate section or as part of your methodology.

34. Page 11, line 3: Why have topography and landcover been aggregated together in one model when all other domains, soil, geology, climate have their own models?

35. Pages 12-14: Sections 3.4 and 3.5 seem to be part of your methodology rather than an explanation of your case study.

36. Page 12, Lines 28-30: Is there one BART tree for each predictor set? Or does each predictor set have multiple BART trees which it can learn from?

37. Page 12, Lines 29:30: I like this approach of having separate models for different types of environmental characteristics.

38. Page 13, Lines 5-9: This description of the benchmark doesn't seem to be about predictor partitioning or data partitioning at all. The theory behind the benchmark estimation should probably be in a method section. Could then show the benchmark model (as a map in the results section and compare it to the 6 BART models and the

original data).

39. Page 13, Lines 5-9: I think one line on how kernel density estimation works would be useful. 40. Page 14, Lines 1-2: sentence doesn't sound right linguistically "how certain a BART model can infer that relationship".

41. Page 14, Lines15-16: If this is the case, is it reasonable to use so many geology predictors. It could unfairly improve the performance of the geology BART model making geology appear to be more important than what it actually is.

43. Pages 14 & 15: Confused as to what the different variances are showing/what you mean.

44. Page 15, Figure 4: Instead of calling models by 1-6. Would be better to call them Aridity, P and Ep, All climate variables, topography, geology, soil. Then reader doesn't need to keep flipping to the appendix to find out what the different model numbers represent. This would be good convention for all other figures as well.

45. Page 15 Lines 4: "regardless of the predictor set, the total predictive variance is always lower than the variance of the benchmark model". Would be good to show the variance of the benchmark model in figure 4.

46. Page 15 Line 12: Why is it surprising that the aridity model improved RMSE the most. Climate is one of the dominant controls for groundwater recharge.

47. Page 16: Would be nice to see maps of RMSE reduction across each of the 6 models. Perhaps this would be needed in supplemental information. This would help understand where the different environmental characteristic types are more or less important.

48. Page 16, Lines 13-18 & Page 18, Figure 7: Figure 7 seems to be a possible example of different conditions. Would be good to have a second figure next to this one which showed what each of the 6 models looked like in comparison to this. Otherwise I'm struggling to understand what the message is.

[Figure]

49. Page 18: I don't think the section title "RMSE labels" is informative as to what this section is about. What does this section actually show?

50. Page 19: Figure 8: • What do the different colours correspond to? • Again, instead of k =1,2,3. . . maybe soil model, Geology model etc would be better. • Code names are not always intuitive "NLCD01_41" "BGEOL_147"- what does that mean? • Tree branch width may be more intuitive if its proportional to the number of watersheds going down each branch. End node impurity is already shown at the bottom of each node.

51. Page 20: I don't think the section title "LPD labels" is informative as to what this section is about. What does this section actually show?

52. Page 21: Figure 9: Similar comments to Figure 8.

53. Page 23 Lines 1-6: I think this suggests that your method of partitioning data, i.e. by their basin, may not have been the best option.

---

## Referee Comment (RC3) · Anonymous Referee #2 · 14 Feb 2019

1. This paper should focus more on process controls.

From what I understand of their nested tree framework, whilst building individual BART models for each domain (climate, geology, soils etc) and evaluating them using CART may help understand where each domain is the dominant control on recharge. Each BART model by itself has limited predictive ability in ungauged watersheds because it is only trained using variables from one domain. If you wanted to use BART models for prediction in ungauged watersheds, perhaps one or an ensemble of BART models which have been trained on all the possible variables would be a better option, as

this would capture the interactions of variables from different domains when predicting recharge.

---

## Author Comment (AC1) · 2 Mar 2019

We thank the Anonymous Reviewer 1 for the constructive comments, which are valuable to improve the quality of this manuscript. Please find our responses in below.

1. We will add the discussion on the lack of temporal coverage as one of the limitation of the case study.

2. We will add the discussion on the lack of coverage over a desirable range of values as one of the limitation of the case study.

[Figure]

3. Agreeing with the reviewer, we acknowledge that some of the findings are specific to the case study, but the generality of the nested tree-based modeling approach is not. In a nutshell, the approach's Bayesian feature sets it apart from other approaches, as the limitation in data accentuates the need to account for uncertainty. The nested structure allows modelers to account for model parameter uncertainty in each individual BART model, and account for conceptual model uncertainty by proposal multiple plausible BART models and comparing them under the nested structure. The nested tree-based modeling approach can help us obtain an informed empirical probability mass function of the plausible BART models (which was exemplified in the case study). This part of the contributions is general, and independent of the case study. The other part of the contributions (including the shift in dominant controlling factor, the pivotal role of soil available water content, etc.) is indeed specific to the case study, and we will try our best to discuss the two parts separately, to reduce confusion. The explanation above will be included in the revised discussion section, and we thank the review for this precious comment.

4. This will be addressed in the same discussion mentioned in the response to comment 3 above.

Comments about the figures:

1. Wording will be changed as suggested.

2. A revised version of the Figure will be provided.

3. The suggested addition will be made.

4. Font size of node numbers will be increased.
* * *

---

## Author Comment (AC2) · 2 Mar 2019

We thank the Anonymous Reviewer 2 for the appreciation of the study, and also for the detailed comments. Please find our responses in below.

General Notes:

1. At the end of introduction, we introduced the two objectives of this study. The first one is proposing an approach that features simultaneous full Bayesian quantification of uncertainty and non-linear regression to model the predictor-response relationship.

[Figure]
The second one is proposing a hypothesis of hierarchical hydrologic similarity and study the key controlling factors of a dynamic hydrologic similarity system.

The two sentences mentioned by the reviewer will be rephrased to avoid confusion.

2. We appreciate the review's comment. However, at this point we intend to keep Sections 3.4 and 3.5 in the case study Section. There are multiple ways to partition the data and multiple metrics with which we can evaluate predictive distributions. Sections 3.4 and 3.5 only introduce our ways that were applied in the case study, and thus are very specific to the case study. A generic study on data partitioning or distribution evaluation is outside the scope of the present study. The materials we put in Section 2 are general and independent of the case study.

To reduce confusion, Section 2 will be revised to be more general, and we will avoid including materials specific to the case study in Section 2.

3. We thank the reviewer for the comment. Data partitioning will be kept in Section 3 and removed from Section 2.

4. We agree with the reviewer. We will remove the Bayesian model averaging Section, but will still briefly mention it to explain how one extra step could be taken to refine the estimates.

5. We thank the reviewer for pointing this out. As we found it difficult to move the explanation of the transformation to Section 3.1 because the climate variables are only explained in Section 3.2, we made the explanation of the transformation its own sub-section, Section 3.2.1.

6. We thank the reviewer for the precious comment, and we agree that it is better to explain the approach in layman terms. As a matter of fact, in the beginning of Section 2 we mentioned that this paper will only provide a brief conceptual introduction to BART, and we provided two excellent studies for readers interested in the details. Explanation of the approach without equations will be added, and Section 2.3 and 2.4
will be revised.

7. Explanation of CART, citation to a paper, and the schematic diagram of a simple example of how BART models are nested under CART will be provided in Section 2.

8. We agree with the reviewer's understanding that in-situ means taken from/at the site/location of interest. Definition of "in-situ" and "ex-situ" will be added when they first appear. The reason the terms are used when we discuss partitioning is because we cannot evaluate accuracy at real ungauged watersheds. Therefore, we partition the data into training set and testing set, and treat the testing set as if they were ungauged watersheds without in-situ data, during the model training phase. With respect to the testing set, the training set provides the ex-situ data (i.e., not from the site/location of interest). The explanation will be added to the manuscript right before we discuss data partitioning, to reduce confusion.

9. We thank the reviewer for making such a suggestion, and we agree that studying the geographic distribution of may provide insights from a different angle. However, in the present study, as discussed in the introduction, we would like to avoid understanding hydrologic similarity with geographic space, and focus more on the predictor space, which can be explored with the nested tree-based approach.

10. We agree with the reviewer that a more intuitive name convention is always desirable. In fact, we tried showing the descriptions of all code-named predictor in the text and in the Figure. However, that lead to unnecessarily lengthy discussion and distorted Figures (in order to fit in the long description of some of the predictors). Thus, we have come to the solution of providing look-up tables. To alleviate the trouble brought by flipping to the tables at the back, we will submit the next draft with tables located near the texts referring to them, and the table will be simplified.

11. We thank the review for making this suggestion. The Figures will be revised for better clarity, and legends of the color coding will be added.

12. There is no comment 12.

13. We thank the reviewer for pointing out the confusing wording. As explained in Section 3, the target response in the case study was not annual recharge itself, nor recharge ratio, but the logit transformed normalized recharge. To reduce confusion and also avoid lengthy text, we will introduce the acronym "LNR" for logit normalized recharge which it first appears, and we will use the term LNR when referring to the target response in the case study.

14. We appreciate this precious comment, which is similar to one of the comments from another reviewer. This is an indication that we did not convey the message clear enough, and we will make corresponding revision for that. Agreeing with the reviewer, we acknowledge that some of the findings are specific to the case study, but the generality of the nested tree-based modeling approach is not. In a nutshell, the approach's Bayesian feature sets it apart from other approaches, as the limitation in data accentuates the need to account for uncertainty. The nested structure allows modelers to account for model parameter uncertainty in each individual BART model, and account for conceptual model uncertainty by proposal multiple plausible BART models and comparing them under the nested structure. The nested tree-based modeling approach can help us obtain an informed empirical probability mass function of the plausible BART models (which was exemplified in the case study). This part of the contributions of the paper is general, and independent of the case study. The other part of the contributions (including the shift in dominant controlling factor, the pivotal role of soil available water content, etc.) is indeed specific to the case study, and we will try our best to discuss the two parts separately, to reduce confusion. The explanation above will be included in the revised discussion section, and we thank the review for this precious comment.

15. This will be added to the discussion mentioned in response 14 above. We will cite a comprehensive study on BART for readers interested in the details of training BART models. Like all models, the fewer data for training the more uncertain the model parameters. Our argument is not that BART is the most accurate model or the

most efficient one in terms of training, but that it offers a Bayesian representation of parameter uncertainty, which we think is of great importance at ungauged watersheds. The arguments above will be included in the revised discussion section.

16. Like the two comments above, we will discuss the generality of the approach, the importance of uncertainty, and the other findings specific to the case study in the revised discussion section.

17. We thank the reviewer for the appreciation.

18. We thank the reviewer for the suggestion. A brief explanation will be added to Section 3.4.2. We agree that more sophisticated division could help us learn something extra. However, as explained in Section 3.4.2, by no means do we expect our partitioning to yield an exhaustive list of all possible sets. We consider the effect of different proposals of plausible BART models (which represents different perspectives of the conceptual understanding of the underlying physics) an interesting follow-up that could be pursued in future studies, but beyond the scope of the present study.

19. We thank the reviewer for the suggestion. Dimension reduction of the data is certainly an interesting way forward. In fact, digging into the geology BART model, we found only a few bedrock types being frequently used as the splitting variables, and the others share a rather uniformly low appearance rate. When doing the case study, we did not have the lithological expertise to aggregate the lithology data ourselves, so we resorted to BART and let the data teach us about the dominant bedrock type. At the early stage of the study, we also tried performing principle component analyses before building BART models, and use the principle components as the predictors. However, we found that this obscured the interpretation of hierarchical similarity and the probability mass function of plausible models, so we turned our attention back to using the predictors as is. Like the response to comment 18 above, we consider the effect of dimension reduction and data aggregation an interesting follow-up that could be pursued in future studies, but beyond the scope of the present study.

20. We agree with the reviewer that the modeler has the opportunity to decide how to partition the data, and agree that a designed partitioning that makes the training and testing samples overlap could improve the robustness of the estimation. The reasons we adopt the MRB-based are listed in Section 3.4.1. To elaborate on reason 1, we would like to avoid training the BART models at the watersheds adjacent to the testing watershed. Adjacent watersheds may share a lot of similarities, and the confounding effects could obscure the results of interest. Reason 2 is a limitation and will be discussed in the revised discussion section.

21. We will elaborate on benchmark model, and will add a reference on kernel density estimation. It is actually quite naïve and does not require any background knowledge; that is why it is used as a benchmark.

22. Like the reviewer suggested it is not necessary, and will be removed.

23. Will be illustrated with a simple example and a schematic diagram.

24. We were explaining how data availability could hinder the application of physically based model. For example, a model of the vadose zone flow may require a water retention curve, which is not always available.

25. Each plausible predictor set corresponds to one BART model.

26. The explanation will be revised.

27. The details of Bayesian model averaging will be removed, but it will still be mentioned in the manuscript.

28. A map will be added.

29. The justification will be moved to the Section where the recharge data are first introduced.

30. The long term variables could compensate for the lack of data on antecedent condition. A detailed discussion will be added to the revised discussion section.

31. We agree that it would be ideal to have the same averaging period. Limited by data availability, we opt to make a working assumption of long-term steady state. This will be added in Section 3.2.

32. Yes. This will be made clear in the revised manuscript.

33. To avoid confusion a new subsection will be added.

34. Answered by the response to comment 18 above.

35. Answered by the response to comment 2 above.

36. One unique predictor set corresponds to one BART model.

37. We thank the reviewer for the appreciation.

38. Yes indeed, the benchmark model does not require predictors at all and is quite naïve and simple. Elaboration on the benchmark will be added in its own subsection to reduce confusion.

39. Instead of a one-line explanation, references will be added.

40. Will be rephrased.

41. Answered by the response to comment 19 above.

42. We could not find comment 42.

43. The algebraic explanation is provided in Section 3.5. Below is the descriptive explanation. From BART, we can obtain a predictive distribution that follows the form of a Gaussian distribution, where both the Gaussian mean and the Gaussian variance are uncertain and are modeled as random variables. What we termed "predictive variance" is the value of that Gaussian variance. Because it's uncertain, we estimated it with the sample median value. What we termed "estimate variance" is the variance of the Gaussian mean, which we estimated with the sample variance of the Gaussian mean.

44. Answered by the response to comment 10 above.

45. It was shown by the red horizontal line in panel (c).

46. Because that BART model only uses two predictors. Before see the results, we thought it would not outperform other models this much on average.

47. Answered by the response to comment 9 above.

48. We thank the reviewer for the suggestion. This Figure is supposed to be an example for the conceptual understanding. Instead of adding another Figure, we will revise the explanation in the manuscript and emphasize the take-away message from this example.

49. Title will be changed to "Nesting by RMSE".

50. Answered by the response to comment 10 above.

51. Title will be changed to "Nesting by LPD".

52. Answered by the response to comment 10 above.

53. We agree that the partitioning was not done perfectly. The reasons for the partitioning are shown the response to comment 20 above.

---

## Author Comment (AC3) · 2 Mar 2019

We thank the Anonymous Reviewer 2 for this insightful suggestion. As suggested, we will revised the manuscript so that the findings regarding the hierarchical similarity will be emphasized more.

---

## Author Response (AR1)

We thank the editor and two anonymous referees for the comments and for giving us the opportunity to improve the presentation of this paper.

Below are our point-by-point replies to the referees' comments, followed by a marked-up manuscript version showing the changes made.

**Responses to comments by anonymous Referee #1:**

1. The discussion on the lack of temporal coverage as one of the limitations has been added as Section 5.3.3, line 5 on page 28 in the revised manuscript.

2. The discussion on the lack of desirable data coverage as one of the limitations has been revised; please see Section 5.3.2, line 28 on page 27 in the revised manuscript. In addition, Section 5.1 provides a detailed discussion on the transferability of the approach, which serves as a basis on which we pose our arguments in Section 5.3.2.

3. We agree with the referee that some of the findings are specific, while some others are general. We added a new discussion in Section 5.1 (line 31, page 25 in the revised manuscript) to discuss the transferability of the approach in details, and put emphasis on what are the innovations in this approach and why those innovations make the approach advantageous at ungauged watersheds.

4. Discussion on the transferability of the approaches has been provided in Section 5.1.

5. Wording in Figure 1 (c) (page 6) has been changed to general notations.

6. A revised Figure of the study area is provided; please see Figure 3 (page 10) in the revised manuscript.

7. The suggested addition has been made; please see Figure 4 (page 12) in the revised manuscript.

8. Please see Figures 9 and 10 (page 22 and 25, respectively) in the revised manuscript; Figures have been redesigned for better clarity and node numbers have been enlarged.

**Responses to comments by anonymous Referee #2:**

1. The introduction has been revised for better clarity of the research objectives; please see line 15 on page 4 in the revised manuscript. The two objectives are proposing a new approach and reveal the key controls of hydrologic similarity for recharge estimation.
   The sentence "…to understand what behavior we can expect in ungauged watersheds" (line 29, page 7 in the revised manuscript) is a statement from a previous study, which they referred to as the ultimate goal of predictions at ungauged basins. We use this statement to support our argument that we have shown why our approach is advantageous, without discussion the superiority of either data-driven or physically based approaches.
   The sentence "The second main objective of this study is revealing the key controlling factors of

a dynamic hydrologic similarity system……" has been removed from Section 2 to avoid confusion. Now the research objectives are all in Section 1, at the end of the introduction.

2. We appreciate the referee's comment. However, at this point we intend to keep Sections 3.4 and 3.5 in the case study Section. There are multiple ways to partition the data and multiple metrics with which we can evaluate predictive distributions. Sections 3.4 and 3.5 only introduce our ways that were applied in the case study, and thus are very specific to the case study. A generic study on data partitioning or distribution evaluation is outside the scope of the present study.

   To reduce confusion, Section 2.3 (line 1 page 8 in the revised manuscript) has been revised as a general description of the nested approach.

3. The description of data partitioning has been removed from Section 2. Now Section 2 only convers the general description of our approach.

4. Bayesian model averaging has been removed from Section 2 as suggested. We still mention it (line 1 page 9) to show that this is a feasible extra step for those who are interested.

5. We have reorganized the materials about normalization in a new subsection, Section 3.2.1 (line 28 page 11).

6. In the beginning of Section 2 (line 24 page 4) we have stated that in this paper we only provide conceptual introduction to BART, and provided two previous studies for those interested in the details. We have revised our explanation (line 6 through 17 on page 5) about the basic concept of BART for better clarity. The number of equations in Section 2 has been reduced to 6 instead of 10.

7. Explanation of CART has been added; please see line 10 through 17 on page 5. An example and a schematic diagram explaining the nesting of BART under CART have been added; please see line 6 through 18 on page 8, as well as Figure 2 in the revised manuscript.

8. Definitions of ex-situ and in-situ data are added when they first appear, line 8 page 1 and line 1 page 1 in the revised manuscript, respectively.

9. We thank the referee for making such a suggestion, and we agree that studying the geographic distribution of may provide insights from a different angle. However, in the present study, as discussed in the introduction, we would like to avoid understanding hydrologic similarity with geographic space, and focus more on the predictor space, which can be explored with the nested tree-based approach.

10. We agree with the reviewer that a more intuitive name convention is always desirable. In fact, we tried showing the descriptions of all code-named predictor in the text and in the Figure. However, that lead to unnecessarily lengthy discussion and distorted Figures (in order to fit in

the long description of some of the predictors). Thus, we have come to the solution of providing look-up tables.

To alleviate the trouble brought by flipping to the tables at the back, we moved Table 5 right next to Figure 9 (please see page 22 in the revised manuscript).

11. Legends have been added in Figures 9 and 10 (page 22 and 25, respectively).

12. There is no comment 12.

13. To reduce confusion and also avoid lengthy text, we introduced the acronym "LNR" for logit normalized recharge (line 2 page 12 in the revised manuscript), and we used the term LNR when referring to the target response in the case study.

14. A detailed discussion about the transferability and the advantageous of the proposed approach has been added in Section 5.1 (line 31 page 25 in the revised manuscript). We emphasize the innovation of our approach on the quantification of uncertainties, which is a general advantage of our approach at ungauged watersheds.

15. A discussion has been added in Section 5.1. We emphasize the innovation of our approach on the quantification of uncertainties, which is a general advantage of our approach at ungauged watersheds. Our argument is not that BART is the most accurate model or the most efficient one in terms of training, but that it offers a Bayesian representation of parameter uncertainty, which we think is of great importance at ungauged watersheds.

We mentioned two studies in Section 2 (line 26 and 27 on page 4 in the revised manuscript) that provide the details of posterior inference statistics with BART, for those interested in the details about how to train BART for prediction purposes.

16. We thank the referee for the comment. We have added Section 5.1 to discuss the general contributions of this study, and Section 5.2 to discuss the contributions that are specific to the case study (which corresponds to the referee's suggestion of process control).

By separating the innovations in the approach and the findings of process control, we hope to separate the two types of contributions of this study, in order to reduce confusion.

17. We thank the referee for the appreciation.

18. We have revised the explanation of predictor partitioning (line 6 through 11 page 15 in the revised manuscript). The very next paragraph (line 12 page 15) explains that by no means do we expect our partitioning to yield an exhaustive list of all possible sets, nor do we expect to include the "best" set.

We consider the effect of different proposals of plausible BART models (which represents different perspectives of the conceptual understanding of the underlying physics) an interesting follow-up that could be pursued in future studies, but beyond the scope of the present study.

19. We thank the reviewer for the suggestion. Dimension reduction of the data is certainly an interesting way forward.

    When doing the case study, we did not have the lithological expertise to aggregate the lithology data ourselves, so we resorted to BART and let the data teach us about the dominant bedrock type. In fact, it turned out that the BART models are capable of identifying a few dominant predictors in a predictor set. We found only a few bedrock types being frequently used as the splitting variables, and the others share a rather uniformly low appearance rate.

    At the early stage of the study, we also tried performing principle component analyses before building BART models, and use the principle components as the predictors. However, we found that this obscured the interpretation of hierarchical similarity and the probability mass function of plausible models, so we turned our attention back to using the predictors as is.

    Like the response to comment 18 above, we consider the effect of dimension reduction, data aggregation, and the variable dominance of different rock types interesting follow-ups that could be pursued in future studies, but beyond the scope of the present study.

20. We agree with the reviewer that the modeler has the opportunity to decide how to partition the data, and agree that a designed partitioning that makes the training and testing samples overlap could improve the robustness of the estimation.

    The reasons we adopt the MRB-based are listed in Section 3.4.1 (line 9 page 15 in the revised manuscript), and this decision is further discussed in the revised Section 5.3.2 (line27 page 27). Also, in the discussion in Section 5.1 we have pointed out that the innovation we emphasis is the Bayesian representation of uncertainty rather than a guaranteed high accuracy of prediction.

21. We have added a new subsection: Section 3.4.3 (line 17 page 16 in the revised manuscript) to explain the benchmark model in details. References on kernel density estimation have also been added in Section 3.4.3.

22. We have removed the description of the structure of this paper, as suggested by the referee. Please see line 1 page 5 in the marked-up manuscript attached below.

23. A schematic diagram and an example have been added. Please see Section 2.3 and Figure 2 on page 8 in the revised manuscript.

24. The sentence has been rephrased for better clarity; please see line 20 page 7 in the revised manuscript. The intention is to explain how data availability could hinder the application of physically based model.

25. Each plausible predictor set corresponds to one BART model. This is also explained in the first paragraph in Section 2.3; please see line 1 through 5 on page 8 in the revised manuscript.

26. Please see line 5 through 23 on page 9 in the revised manuscript for the revised explanation. Two examples have been added.

27. Bayesian model averaging has been removed as suggested.

28. A map has been added; please see Figure 3 on page 10 in the revised manuscript.

29. As suggested, justification has been moved to Section 3.1 (line 1 through 9 on page 11 in the revised manuscript).

30. The long term variables could compensate for the lack of data on antecedent condition. A detailed discussion will be added to the revised discussion section. Please see line 24 through 26 on page 11, as well as Section 5.3.3 (page 28) in the revised manuscript.

31. A discussion has been added in Section 5.3.3, page 28 in the revised manuscript.

32. Answered by the response to comment 13.

33. Answered by the response to comment 5.

34. Answered by the response to comment 18.

35. Answered by the response to comment 2.

36. Answered by the response to comment 25.

37. We thank the referee for the appreciation.

38. Answered by the response to comment 21. Yes, like the referee said, the benchmark model does not require predictors at all and is quite naïve and simple.

39. Answered by the response to comment 21. References on kernel density estimation has been added.

40. The sentence has been rephrased as suggested; please see line 9 on page 20 in the marked-up manuscript attached below.

41. Answered by the response to comment 19.

42. There is not comment 42.

43. The algebraic explanation is provided in Section 3.5. Below is the descriptive explanation. From BART, we can obtain a predictive distribution that follows the form of a Gaussian distribution, where both the Gaussian mean and the Gaussian variance are uncertain and are modeled as random variables.
What we termed "predictive variance" is the value of that Gaussian variance. Because it's uncertain, we estimated it with the sample median value.

What we termed "estimate variance" is the variance of the Gaussian mean, which we estimated with the sample variance of the Gaussian mean.

44. Answered by the response to comment 10.

45. Please see the red horizontal line and the caption of Figure 5, on page 18 in the revised manuscript.

46. It was surprising because it was unexpected that the model with only two predictors outperformed the other models in general. It not only outperformed models with non-climate predictors, but also outperformed models with other climate predictors.

47. Answered by the response to comment 9.

48. We thank the reviewer for the suggestion. This Figure is supposed to be an example for the conceptual understanding rather than an actual case.
Instead of adding another Figure, we have revised the explanation (line 17 through 33 on page 20 in the revised manuscript). In particular, please see line 27 through 30 on page 20 in the revised manuscript, where we mention why the phenomenon in Figure 8 in the revised manuscript is possible.

49. Please see page 21 in the revised manuscript; the title has been rephrased.

50. Answered by the response to comment 10.

51. Please see page 24 in the revised manuscript; the title has been rephrased.

52. Answered by the response to comment 10.

53. We have added explicit acknowledgment that the data partitioning in the case study is not the best partitioning; please see Section 5.3.2 on page 27 in the revised manuscript. Section 5.3.2 provides a discussion on the limitations due to the partitioning, as well as the reason we kept this partitioning method.

**Responses to the additional comment by anonymous Referee #2:**

We thank the referee for this insightful suggestion. We have revised the discussion section to discuss the general contributions and the specific contributions of this study separately. In Section 5.1 we discuss the innovations in our approach, and emphasize the advantage of the quantification of the parameter uncertainty as well as the model structure uncertainty. In Section 5.2 we discuss the findings specific to the case study. This is the discussion on "process controls" as suggested by the referee.

[revised manuscript text omitted]
 Deciduous Evergreen Mixed Dwarf shrub Shrub/scrub Grassland Sedge Lichens Moss Pasture/hay Crops Woody wetland Emergent herbaceous wetland

Soil property predictors. **Soil property Unit Statistics\*** Calcium carbonate equivalent % Lower/higher bounds Cation exchange capacity cmolc / kg Lower/higher bounds Depth to the seasonally high water table m Average and Lower/higher bounds Soil thickness m Lower/higher bounds Hydrologic soil group classification % Average Soil erodibility factor dimensionless Average Permeability m / $h^{-1}$ Average and Lower/higher bounds Available water content fraction Average and Lower/higher bounds Bulk density g / $cm^3$ Average and Lower/higher bounds Organic matter content % Average and Lower/higher bounds Clay soil content % Average and Lower/higher bounds Silt soil content % Average Sand soil content % Average Percent finer Than nos.4, 10, and 200 sieve % Average and Lower/higher bounds

Table of the six different predictor sets. **kpredictors included** 1$\bar{\phi}$ and $\phi$ 2 2$\bar{P}$, $P$, and $E_p$ 3 3All climate predictors: $\bar{P}$, $P$, $E_p$, $\bar{\phi}$ and $\phi$ 5 4Topography and land cover predictors 20 5Soil predictors 48 6Geology predictors 206

Reference list of the splitting variables in Fig. 9 and Fig. 10 **Node number Splitting variable Node Number Splitting variable** 1 Average available water content(AWCAVE)1 Average available water content(AWCAVE)3 Long term average aridity index(ARID_IDX)3 Long term average aridity index(ARID_IDX)4 % area of Paragneiss and Schist bedrock(BGEOL_147)4 Precipitation in 2002(PPT02MEAN)5 Average slope(SLP_DEG)6 Precipitation in 2002(PPT02MEAN)7 Precipitation in 2002(PPT02MEA Precipitation in 2002(PPT02MEAN)10 % area of Deciduous Forest(NLCD01_41)15 Aridity index in 2002 (ARID_IDX02)17 Precipitation in 2002(PPT02MEAN)19 Average slope(SLP_DEG)